astrophysics

planet formation, star formation, extrasolar planets, Solar System

**Author for correspondence:**
Richard J. Parker
e-mail: R.Parker@sheffield.ac.uk

# The birth environment of planetary systems

## Richard J. Parker

Royal Society Dorothy Hodgkin Fellow, Department of Physics and Astronomy, The University of Sheffield, Hicks Building, Hounsfield Road, Sheffield S3 7RH, UK

RJP, 0000-0002-1474-7848

Star and planet formation are inextricably linked. In the earliest phases of the collapse of a protostar, a disc forms around the young star and such discs are observed for the first several million years of a star's life. It is within these circumstellar, or protoplanetary, discs that the first stages of planet formation occur. Recent observations from the Atacama large millimetre array (ALMA) suggest that planet formation may already be underway after only 1 Myr of a star's life. However, stars do not form in isolation; they form from the collapse and fragmentation of giant molecular clouds several parsecs in size. This results in young stars forming in groups—often referred to as 'clusters'. In these star-forming regions, the stellar density is much higher than the location of the Sun and other stars in the Galactic disc that host exoplanets. As such, the environment where stars form has the potential to influence the planet formation process. In star-forming regions, protoplanetary discs can be truncated or destroyed by interactions with passing stars, as well as photoevaporation from the radiation fields of very massive stars. Once formed, the planets themselves can have their orbits altered by dynamical encounters—either directly from passing stars or through secondary effects such as the Kozai–Lidov mechanism. In this contribution, I review the different processes that can affect planet formation and stability in star-forming regions. I discuss each process in light of the typical range of stellar densities observed for star-forming regions. I finish by discussing these effects in the context of theories for the birth environment of the Solar System.

## 1. Introduction

When I was asked to contribute a review article to this Royal Society volume, one topic immediately sprung to mind. It is something that, quite obviously as I am authoring this review, I find fascinating, but I have yet to find either a professional astronomer, an amateur astronomer or a layperson who is not fascinated by the question of whether our Solar System is typical among the thousands of planetary systems orbiting other stars. In a sense, the question can be conceptually quite straightforward; all we need to do is to

work out what the typical conditions are for star formation, and then model these conditions as planets are forming(!). In reality, this is an extremely complicated thing to work out, and this review article aims to give the reader a flavour of the many different strands of research that make up this problem.

During my PhD studies (2007–2010) I worked on computing the rate of dynamical encounters in star-forming regions, and how these encounters could disrupt primordial binary star systems. One of the underlying assumptions of this work (an assumption which I now consider to be at best too simplistic, and at worst completely wrong) was the idea that all star-forming regions had very high stellar densities (greater than $1000 \, \mathrm{M_\odot} \, \mathrm{pc^{-3}}$, which is four orders of magnitude higher than the stellar density in the local Milky Way Galaxy near our Sun). This assumption was born of its time; the previous 20 years of observations of young stars had shown them to be grouped together in their birthplaces—giant molecular clouds (GMCs)—and although one can measure the present-day density of a star-forming region, at that time it was extremely difficult to infer, or guess at the *initial* density of a star-forming region, other than the present-day density is probably a lower limit to the initial density.

Whether this assumption is right, wrong, or as is the case with most things, somewhere in the middle, as a PhD student I wondered what effect such a high stellar density would have on planetary systems. I wondered whether they would be able to form at all, and if they were able to form, what the long-term effects of being born in a dense environment would be on such planets. Whether by fluke or design, my boss in my first post-doctoral position put me in the same office as an expert on directly imaging exoplanets. One day my office-mate asked 'Do you ever put planets in those simulations of yours?', and that conversation eventually led to Parker & Quanz 2012 [1], the first piece of work I did that was independent of my PhD supervisor.

I remember the literature search for that paper being a Herculean slog; I rapidly realized many people had done similar work before, but in the late 1990s and early 2000s the field had moved so rapidly that a comprehensive review article on the topic was lacking. I subsequently started working on the effects of the star-forming environment on the early Solar System (on which Fred Adams had written an excellent review back in 2010 [2]), but then struggled to assimilate and digest the vast literature on external photoevaporation of protoplanetary discs; something that turns out to be relevant for both Solar System formation *and* the formation of planets in star-forming regions in general.

Later, when I was fortunate enough to be supervising other people's PhD theses, these students would invariably say the same thing: why isn't there a review article broadly covering the physics of the birth environment of planetary systems? Just as I had in my PhD, they were missing a starting place to learn not just about star formation, but star cluster dynamics, planetary dynamics and protoplanetary disc destruction.

In recent years, huge strides have been made in all of these areas. The initial conditions for *N*-body simulations of star-forming regions now try to mimic the filamentary and substructured appearance of observed star-forming regions, whereas the early simulations were limited to modelling smooth, spherically symmetrical star clusters. Graphical processor units (GPUs) have enabled multi-planet systems to be modelled within *N*-body simulations of star clusters, which is making these studies more realistic than ever before. At the same time, state-of-the-art modelling of both photoevaporation and the internal evolution of discs—again, within global *N*-body simulations of star-forming regions—is completely changing our understanding of planet formation in environments where massive stars are present.

My intention with this article is not to cover everything, but to give an overview that is accessible to Masters and PhD students, as well as people who may not work directly in this field. I begin by giving an overview of star formation to try to qualify the density of star-forming regions where planet-host stars form, and how these regions may disperse into the Galactic disc. I then discuss the effects of the star-forming environment on protoplanetary discs, before moving to a discussion on the effects on fully formed planets. I finish with a short review of the birth environment of the Solar System.

If you only take one thing away from this review, at least consider that star and planet formation are inextricably intertwined, and cannot be treated as isolated events. And also realize that even though this is a review, and it has been refereed, this article will still be biased, if only from the selection of material I have chosen to include.

## 2. Star formation

When discussing the likely birth environments of planetary systems, it is first useful to consider the formation environment of the planets' host stars. Stars like the Sun do not form in isolation, but rather in the company of many other young stars. Therefore, the regions in which stars form are often called

'clusters', although this term is used interchangeably with 'groups', 'star-forming regions' and 'associations'. Various definitions exist for each of these [3,4]; from herein I will use the catch-all term 'star-forming region' to describe any collection of young (less than 10 Myr) stars. I will limit the use of 'cluster' to describe a gravitationally bound set of stars, and an 'association' to describe a gravitationally unbound set of stars [5].

Observations with the *Spitzer* and *Herschel* space telescopes reveal young stars to be deeply embedded in the GMCs from which they form, and it is thought that this naturally explains the clustering, or grouping, of young stars with other stellar objects. GMCs are predominantly composed of molecular hydrogen, and have temperatures of only several tens of K. Parts of the GMCs are expected to collapse and then fragment, with the mass-scale for each fragment (the Jeans mass) around $1 \, M_\odot$ [6,7] (which corresponds to the peak of the distribution of pre-stellar cores, [8]) and the size of the fragment around 0.1 pc, which corresponds to the typical radius of a core.

These pre-stellar cores undergo collapse to pre-main sequence stars (objects with stellar densities that have not commenced hydrogen fusion in their cores) and also further fragmentation, so that an individual core can form a single star or a multiple star system [9] and thus have masses straddling the peak of the initial mass function (IMF)—between 0.2 and $0.5 \, M_\odot$. Multiple star systems are small collections of stars—either binary, triple, quadruple (or even higher-order) systems—that orbit a common centre of mass. Multiple systems—especially triple, quadruple or higher—are not always stable for the lifetime of a star, but observations show that almost half of all Solar-mass ($0.8 < m/M_\odot < 1.2$) stars are in binary, triple or quadruple systems [10,11]. Indeed, because both dynamical decay (where an unstable multiple system loses members) [12,13] and direct destruction from encounters [14–16] destroy multiple systems and create more single stars, the primordial, or birth binary fraction, may be even higher than 50%.

The distribution of individual stellar masses formed from the collapse and fragmentation of a GMC usually follows a lognormal distribution at low masses ($0.1$–$1 \, M_\odot$, with a peak between 0.2 and $0.5 \, M_\odot$), and above $1 \, M_\odot$ it follows a power-law slope of the form

$$\frac{\mathrm{d}N}{\mathrm{d}m} \propto m^{-2.35}, \qquad (2.1)$$

which is sometimes referred to as the 'Salpeter' slope after the seminal paper by Salpeter [17], who was the first to measure the mass function of stars. Most formulations of the stellar mass distribution (the initial mass function) agree that low-mass stars are inherently more common than higher mass stars [17–22], predominantly because of the steep slope of the mass function.

The IMF tells us that on average, the more stars in a star-forming region, the more likely it is that the region will contain massive (greater than $10 \, M_\odot$) stars. The relation between the total mass of a star-forming region and the most massive star that is able to form has been the subject of much debate [23–34], with some authors claiming that a star-forming region must have a mass higher than some threshold (typically higher than approx. $500 \, M_\odot$, [30]) in order to form massive stars (with that threshold being significantly higher than the mass of the massive star(s)). However, several examples of low-mass (approx. $100 \, M_\odot$) star-forming regions are known to contain massive stars, including $\sigma$ Orionis and $\gamma^2$ Velorum.

Massive stars have both a positive and a detrimental effect on planet formation, as we will see in §§3 and 5, and on average the presence of massive stars implies a significant number of lower-mass stars, due to the form of the IMF. This then factors into an important discussion on whether stars form in regions with a typical density (number, or summed mass of stars per unit volume), or range of densities. If stars form within a fundamental length scale, then the more stars there are (i.e. the more massive the GMC), the higher the stellar density is likely to be. By way of an example, if a region contains a few hundred stars within a 1 cubic pc volume, its density is of order $100 \, M_\odot \, \mathrm{pc}^{-3}$. If a region contains several thousand stars within the same 1 cubic pc, the density is a factor of ten higher ($1000 \, M_\odot \, \mathrm{pc}^{-3}$).

It should be emphasized that this is the *average* density within a star-forming region; even a region with a spherical stellar distribution will contain a range of stellar densities. So, for the following discussion the quoted stellar densities are the average (median) densities, but a planetary system may encounter different stellar densities within the same star-forming region, depending on the length of time it spends in that region. For dynamically old regions, a planetary system has probably experienced the full range of stellar densities. This subtle point becomes important when considering the effects on protoplanetary discs, which tend to have shorter or comparable lifetimes to the dynamical evolutionary timescales of star-forming regions.

The density of a star-forming region then sets the level of perturbation a planetary system can expect. There are essentially three density regimes that pose a threat to planetary system formation and evolution. In the presence of massive stars, far ultraviolet (FUV) and extreme ultraviolet (EUV) radiation can affect protoplanetary discs at stellar densities as low as $10 \, M_\odot \, \mathrm{pc}^{-3}$. At stellar densities greater than $100 M_\odot \, \mathrm{pc}^{-3}$,

planetary orbits can be altered (which can include changes to the orbital eccentricity, inclination and semimajor axis of individual planets). At higher densities still (greater than $1000\,M_\odot\,pc^{-3}$) protoplanetary discs can be physically truncated by encounters. (We will discuss each of these processes in more detail in the following sections.)

The density regimes quoted above assume that most star-forming regions are short-lived (less than 10 Myr) entities. One of the most important issues when considering the density of a star-forming region is that the present-day (observed) density of a star-forming region may be significantly lower than the birth, or maximum density [35]. Star-forming regions are collisional systems, and strive to reach equilibrium via dynamical relaxation, which has the effect of causing the region to expand. Depending on the initial density, this expansion may cause a decrease in the stellar density by several orders of magnitude. It is, therefore, impossible to pinpoint the maximum density by simply observing the current density at a given time, and extra information is required.

Stars appear to form in filamentary structures within GMCs [36,37], and so a star-forming region that has not undergone significant dynamical evolution is expected to exhibit a large degree of spatial and kinematic substructure [38–41]. This substructure is progressively wiped out as a star-forming region undergoes relaxation [42,43], meaning that we can use the amount of substructure as a dynamical clock. This has proven effective at predicting the initial density of several observed star-forming regions, and by extension, predicting the amount of perturbations planetary systems in those regions would expect to experience. In general, most star-forming regions are consistent with having initial densities of *at least* $100\,M_\odot\,pc^{-3}$, which would cause some disruption to the orbits of planetary systems, and cause significant mass-loss to the protoplanetary disc if the region contains massive stars.

When considering the environment of either very young planets (less than 10 Myr) or protoplanetary or debris disc host stars, the inferred birth density from nearby star-forming regions is probably valid. However, many of the exoplanet host stars are of a similar age to the Sun and therefore formed at redshifts approaching $z \sim 1$, which coincides with a known period of intense star formation in the Milky Way, betrayed by a high rate of star formation at this time [44]. Environments where the star formation rate is higher than the present average in the Milky Way disc are characterized by very dense star-forming regions [45,46], and so similar conditions in the Galaxy when the Sun and the other exoplanet host stars were born suggests that the environment for planet formation may have been much more hostile [47].

A further uncertainty in assessing the birth environment of planetary systems is that we do not yet understand how star-forming regions disperse into the Galactic field, and whether a certain type of star-forming region (in terms of some property such as total stellar mass, or density, or overall bulk motion) is more likely to dissolve into the disc than any other. Part of the issue is that we do not know the dominant mechanism by which star-forming regions disperse. The most commonly cited mechanism is called 'residual gas expulsion', and uses the fact that star formation is an inherently inefficient process—only around 30% of the mass of a GMC is converted into stars, leaving a significant gravitational potential in the form of gas. As the stars reach the main sequence, the combined effect of their stellar winds, coupled with the effects of the first supernovae, causes the rapid removal of the remaining gas and has the effect of unbinding the star-forming region, causing it to expand and disperse [48–54].

This gas expulsion scenario was initially quite successful at explaining the observed properties of star clusters, especially the apparently supervirial motion of stars in many clusters, and the significant decrease in clusters at ages older than 10 Myr [52,55]. However, subsequent studies showed that the supervirial motion was probably due to a mis-interpretation of the velocity dispersion in these clusters—orbital motion from binary stars has the effect of inflating the velocity dispersion if not accounted for, making the stars in the clusters appear to be moving faster than they are [56]. Furthermore, numerical simulations of star formation in GMCs have shown that the *local* star formation efficiency is quite high, i.e. the locations where groups of stars form tend to be gas-poor—removing this extra mass therefore has a negligible effect on the gravitational potential of the stars in question [57].

The main agent for dispersing star-forming regions may be the tidal field of the Galaxy itself. Each star-forming region has a tidal radius where the gravitational influence of the Galaxy exceeds that of the star-forming region (this is analogous to the Hill sphere in planetary dynamics, or the Roche lobe in contact binary stars). For star-forming regions, the boundary between the gravitational influence of the Galaxy and the gravitational influence of the star-forming region is called the Jacobi radius, $r_J$, and is a function of the position of the star-forming region in the Galaxy, $R_G$, the mass of the Galaxy $M_G$ and the mass of the star-forming region, $M_c$:

$$r_J = R_G \left(\frac{M_c}{M_G}\right)^{1/3}. \tag{2.2}$$

Taking the mass of the Milky Way as $M_G = 10^{12}\,M_\odot$, the Galactocentric radius of the Sun as $R_G = 8\,\text{kpc}$ and the mass of a cluster as $1000\,M_\odot$, then in this case $r_J = 8\,\text{pc}$. Stars ejected from dense clusters typically move at velocities more than $10\,\text{pc}\,\text{Myr}^{-1}$, and so we would expect stars to quickly cross the Jacobi radius. Furthermore, as star-forming regions expand due to dynamical relaxation, eventually stars pass the Jacobi radius and the tidal field of the Galaxy dominates. The star-forming region formally loses mass [58], and therefore its Jacobi radius decreases. This decrease in Jacobi radius then makes the star-forming region even more susceptible to losing stars, and the process repeats until presumably the star-forming region completely disperses. It is not clear how long this process takes, nor how it depends on the initial properties of the star-forming region. However, it is clear that the more massive and compact a region is, the more likely it is to remain a bound entity for a significant fraction of the age of the Universe.

An often overlooked aspect of star formation, but one that is fundamental to our understanding of the Sun's place in the cosmos, is the fact that the majority of stars form as binary, or higher order (e.g. triple or quadruple) systems. In the local Solar neighbourhood, around 50% of stars are in binary systems [11]. In nearby star-forming regions, the fraction of stars in binary systems is either the same, or higher than in the field [59], with some authors arguing that *all* stars form in some sort of multiple system [14,60–62].

It is not clear how much the binarity of stars hinders planet formation and evolution. In a given binary system, there will be some region of the orbit where planets are unstable, and this has been quantified to some extent by numerical simulations [63]. If a planet is on a 'satellite', or S-type orbit (i.e. where the planet orbits one of the two stars), then there will be some maximum semimajor axis beyond which the orbit of the planet is unstable. Similarly, if a planet is on a 'planet', or P-type orbit (i.e. it orbits *both stars*), then there is a minimum semimajor axis, below which the planet is unstable.

Observations have established that planets, and protoplanetary discs, readily form and survive around binary systems [64,65]. However, in a star–forming environment, binaries have the extra complication that they present a larger cross section for encounters with other stars, and therefore an interaction that could subsequently destabilize the planetary system is more likely than around a single star [66].

# 3. Destruction of protoplanetary discs

Some of the very first attempts to describe star formation by Laplace and Kant recognized that conservation of angular momentum would result in a circumstellar disc around stars as they are forming. These discs, composed of dust and gas, are now known to form planets around stars as they themselves are forming.

This particular field has been completely revolutionized by observations with the Atacama large millimetre array (ALMA), an interferometer located in the high and dry desert in Chile. One of the first images to come from ALMA was the now famous HL Tau disc, a disc surrounding a young, nearby star in the Taurus star-forming region [67]. Further observational studies, in particular, the Disc Substructures at High Angular Resolution Project (DSHARP) [68], have shown that discs around young stars display features than can be attributed to a change in disc composition [69–72] as a function of radius, or the presence of fledgling planets within those discs [73–77] (although the latter scenario is still hotly debated [78,79]).

The presence of a protoplanetary disc was previously inferred by examining the spectral energy distribution (SED) of a star, which should display very little deviation from a black-body spectrum. However, pre-main sequence stars often have a strong excess in the SED at infrared wavelengths [80–82], with the only viable explanation being that the star is surrounded by a disc composed of micrometre-sized dust grains, which absorb light from the host star and then re-emit this light at redder wavelengths [83–85]. Some estimates find that the disc mass can be up to 10% of the host star's mass. A lower limit for the mass of the Sun's protoplanetary disc can be estimated by adding up the material in the Solar System (planets, dwarf planets, asteroids) to determine the 'Minimum Mass Solar Nebula', which is at least 1% of the Sun's mass [86,87].

Pre-main sequence stars that display infrared excesses indicative of a disc are almost exclusively found in star-forming regions, and there is evidence of a trend of decreasing numbers of discs with stellar age, such that very young (approx. 1 Myr) stars almost all have discs, whereas at 5 Myr almost no stars have discs [88,89]. This observation is interpreted as either planets form very quickly, or the majority of discs are destroyed (or both).

In star-forming regions, the natural assumption is that many of the discs are destroyed, either by encounters with passing stars, or by the radiation fields from stars more massive than $10\,M_\odot$. The efficacy of both scenarios depends on the stellar density of the star-forming region, and we discuss both

below. (Note that discs can also be depleted by accretion on to the central star, photoevaporation from the central star, and planet formation itself. For the remainder of this review, we take 'photoevaporation' to be from the external radiation fields of massive stars, rather than due to the host star.)

## 3.1. Truncation from fly-bys

Protoplanetary discs can in theory be destroyed or truncated by an encounter with a passing star in a dense stellar environment. Early simulations [90] showed that material would be removed from a disc up to one-third of the distance of the encounter. For example, if the closest approach of a star were 300 AU, then the disc would be truncated to a radius of 100 AU [91].

Further simulations showed that an encounter with the disc truncates the outer edge and also steepens the density profile of the remaining material in the disc [92], and that prograde, co-planar encounters are the most destructive to the disc [93]. Material from the disc can also be transferred onto an orbit around the intruding star [94].

All of these early simulations were essentially isolated systems, with the disc constructed in a hydrodynamical simulation with a simulated encounter with a single, passing star. In reality, a star may undergo multiple interactions in its birth environment. However, current limitations in computing power mean that it isn't possible to self-consistently model discs to the required resolution within $N$-body simulations of realistic star-forming regions (though some attempts at modelling discs in regions where the number of discs is low and the initial conditions of the regions are fairly basic have been made [95]).

The usual approach to quantifying disc destruction in star-forming regions is to apply an after-the-fact post-processing analysis to the $N$-body simulations. The simulations are run without discs, and then discs are 'assigned' to stars and modified according to the encounter history of the individual star. As such, the discs are not real and their gravitational influence (which can be significant if their masses are approx. 10% of the host star) is ignored, as are the possible effects of gravitational focusing during encounters. However, the advantage of this method is that the disc destruction recipes can be easily applied to simulations of an entire star-forming region, and significant amounts of work have been done in this area [96–102].

Using the approach of post-processing in full $N$-body simulations of star-forming regions, most authors tend to find that only the most dense star-forming regions (greater than $10^3 \, M_\odot \, \mathrm{pc}^{-3}$) lead to significant disc truncation. In the vicinity of the Sun (within the nearest 500 pc), the Orion Nebula Cluster (ONC) is probably the only star-forming region that may have been this dense [35]. However, the larger Orion star-forming region makes up a significant fraction of the nearest young stars to the Sun [103] and so many studies have focused on disc destruction in this star-forming region [96,99,100].

For ONC-like clusters, research has found that discs with radii larger than 500 AU are affected by truncations due to the star-forming environment [99], which is consistent with the observation that 200 AU-sized discs are common [104]. In simulations of the more distant (1740 pc) Eagle Nebula, a.k.a. NGC 6611, which is thought to have a higher stellar density than the ONC, the disc sizes are cut down on average to roughly 100 AU. A potentially observable test of these simulations (and others like them) would be that the discs in the centre of NGC 6611 should be on average approximately 20 AU and therefore considerably smaller than those in the ONC.

These simulations are of very dense star-forming regions, where a high degree of dynamical relaxation, involving encounters and mixing between disc-hosting stars, would be expected to affect the discs in equal measure. The simulations find that on the outskirts of clusters, discs above a certain radius tend to have a large size, but there is no dependence on the star's location if the disc radius is less than 100 AU.

To date, observations of star-forming regions indicate that the properties of discs—particularly their radii, which tend to be smaller—are different in the most dense star-forming regions compared to lower-density regions [102,105–107], but is unclear whether this is due to truncation by dynamical encounters, or external photoevaporation, which we discuss in the next section.

An interesting additional problem is that the disc can in principle grow in mass in a star-forming region by accreting gas from the surrounding medium [108–110] (recall that star-formation is inefficient, and young stars are almost always embedded in their natal GMCs, so the reservoir of gas from which stars can accrete is plentiful). Interaction with the background gas can be important and recent results indicate that face-on accretion dominates over mass loss from stellar encounters at low densities [111].

Once the protoplanetary discs have evolved to form planetessimals, they are referred to as 'debris discs'. These discs become even more robust to the environment, with simulations finding that debris discs are only depleted in star-forming regions with stellar densities exceeding $2 \times 10^4 \, M_\odot \, \mathrm{pc}^{-3}$ [112].

## 3.2. Photoevaporation

Some of the first images taken with the Hubble Space Telescope following the much publicized optics correction carried out by space shuttle astronauts were of the Orion 'proplyds'—disc-like structures observed around pre-main sequence stars in the vicinity of the Orion Nebula Cluster [113–116]. Not only were these images some of the first direct detections of protoplanetary discs around young stars, the images also indicated that the star formation environment plays an important role in shaping the planetary systems that subsequently form around young stars.

The 'proplyds' were discs, or more specifically ionization fronts of material being liberated from the discs around young stars in very strong radiation fields, and the term proplyd—a contraction of 'PROtoPLanetarY DiSc'—is usually nowadays only reserved for discs that are being externally irradiated.

The relatively close-by (approx. 400 pc, [117,118]) Orion Nebula Cluster is the only location where these irradiated discs are observed in detail, partly because very few closer star-forming regions contain stars more massive than approximately $5\,M_\odot$ (recall that this is probably simply due to a statistical sampling argument—nearby star-forming regions tend to be low-mass, and so will not form such massive stars).

Stars more massive than $5\,M_\odot$ emit highly energetic photons in both the FUV and EUV regions of the spectrum. The typical energies of individual photons are in the range $5 < h\nu < 13.6$ eV for FUV radiation, whereas the photons from EUV radiation have energies exceeding 13.6 eV. These two radiation regimes constitute a significant proportion of the luminosity from massive stars. For example, collating stellar atmosphere models, a $20\,M_\odot$ star produces an FUV luminosity, $L_{\mathrm{FUV}} \simeq 10^{38}$ erg s$^{-1}$, and an EUV luminosity $L_{\mathrm{EUV}} \simeq 10^{37}$ erg s$^{-1}$ [119]. (Note that $1$ erg $= 1 \times 10^{-7}$ J.) The EUV luminosity exceeds the FUV luminosity only at stellar masses above $60\,M_\odot$.

To determine how much of this radiation is received by a disc-hosting star, we must convert the luminosity into a flux, usually expressed in units of erg s$^{-1}$ cm$^{-2}$. For example, in a typical star-forming region a disc-hosting star may be around 0.5 pc from an ionizing massive star producing FUV and EUV radiation. As $0.5$ pc $= 1.54 \times 10^{18}$ cm, the EUV flux received from a star 0.5 pc from a $20\,M_\odot$ star will be

$$F_{\mathrm{EUV}} = \frac{L_{\mathrm{EUV}}}{4\pi d^2}. \tag{3.1}$$

If $d = 1.54 \times 10^{18}$ cm and $L_{\mathrm{EUV}} = 10^{37}$ erg s$^{-1}$, then $F_{\mathrm{EUV}} = 0.3$ erg s$^{-1}$ cm$^{-2}$. As the FUV luminosity for a $20\,M_\odot$ star is a factor of ten higher, we have

$$F_{\mathrm{FUV}} = \frac{L_{\mathrm{FUV}}}{4\pi d^2}, \tag{3.2}$$

where $L_{\mathrm{FUV}} = 10^{38}$ erg s$^{-1}$ and $d = 1.54 \times 10^{18}$ cm, as before. This gives $F_{\mathrm{FUV}} = 3.34$ erg s$^{-1}$ cm$^{-2}$, but the FUV flux is usually expressed in terms of the background FUV flux in the interstellar medium, referred to as the 'Habing unit' [120], or the '$G_0$ field', where 1 Habing unit is $1\,G_0 = 1.6 \times 10^{-3}$ erg s$^{-1}$, meaning that the FUV flux in this particular scenario is $F_{\mathrm{FUV}} = 2090\,G_0$, i.e. more than 2000 times the background ISM radiation field.

When FUV radiation is incident on a disc, it heats the material in the regions where the surface density of the particles is lowest, which tends to be on the edges of the disc [121]. The thermal pressure exerted causes a wind to be launched from the edge of the disc, allowing gaseous material to escape from the disc [122–124]. This planar geometry means that the process can be effectively modelled as a one- or two-dimensional system. The initial calculations of mass-loss in discs due to incident FUV radiation [125] demonstrated that dust particles could be entrained in the wind from the disc; however, more recent calculations using the `FRIED` grid [126] show that—while the gas component of the disc is readily evaporated—the dust content is largely retained.

The first calculations often assumed that the EUV radiation would dominate the mass-loss from discs. However, in practice, it appears that the wind launched by the incident FUV radiation creates an ionization front around the disc, which shields the disc from further destruction from EUV radiation. The mass-loss due to FUV radiation appears to be extremely high in star-forming regions that contain any massive stars [127]. Even at distances commensurate with those in extended OB associations (several pc), the $G_0$ field from a $20\,M_\odot$ star is still five times that of the field in the ISM.

The most recent calculations (the `FRIED` grid) use the $G_0$ field as an input, as well as the disc mass $M_{\mathrm{disc}}$ and disc radius $r_{\mathrm{disc}}$ to determine the mass-loss rate. As an example, for a disc with initial mass $M_{\mathrm{disc}} = 0.1\,M_\odot$ (a reasonably massive disc), initial radius $r_{\mathrm{disc}} = 100$ AU, an FUV field of $G_0 = 1000$

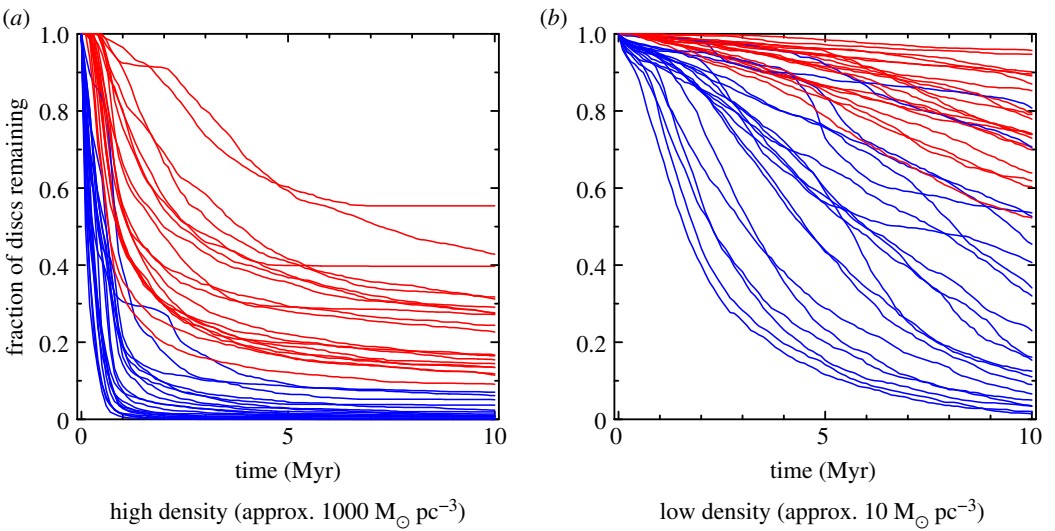

**Figure 1.** Destruction of protoplanetary discs due to external photoevaporation from the radiation fields from massive stars. In (a,b), we show the fraction of discs that survive over time in 20 realizations of the same simulated star-forming region, where the blue lines are discs with initial radii of 100 AU, and the red lines are discs with initial radii of 10 AU. We show two different stellar density regimes: (a) for star-forming regions with initial densities of approximately 1000 M$_\odot$ pc$^{-3}$ (thought to be a reasonable approximation for the initial density of the Orion Nebula Cluster), and (b) for star-forming regions with initial densities of approximately 10 M$_\odot$ pc$^{-3}$ (thought to be a reasonable approximation for the initial density of the Cyg OB2 stellar association).

would completely destroy the gas component of the disc within just 0.1 Myr. However, if the same disc were much more compact (10 AU) then very little mass ($1 \times 10^{-4}$ M$_\odot$) is lost in 1 Myr.

Several authors have performed simulations of the evolution of star-forming regions to quantify the mass-loss due to both FUV and EUV radiation over the first 10 Myr of a star-forming region's lifetime. A detailed set of simulations using the FRIED grid is currently in preparation, but a rough estimate of the mass-loss can be made using the following prescriptions [128], which implements the effects of both FUV and EUV radiation [122–124]. In these models, the FUV field around a massive star is effective out to a radius of around 0.3 pc, beyond which it does not contribute to the mass loss due to photoevaporation. Within this 0.3 pc radius, the mass loss due to FUV radiation (photon energies $hv < 13.6$ eV) is dependent only on the radius of the disc $r_d$ and is given by

$$\dot{M}_{\text{FUV}} \simeq 2 \times 10^{-9} r_d \text{ M}_\odot \text{ yr}^{-1}. \tag{3.3}$$

The disc also loses mass due to EUV radiation (photon energies $hv > 13.6$ eV), and this does depend on the distance from the massive star(s) $d$ as well as the radius

$$\dot{M}_{\text{EUV}} \simeq 8 \times 10^{-12} r_d^{3/2} \sqrt{\frac{\Phi_i}{d^2}} \text{M}_\odot \text{ yr}^{-1}. \tag{3.4}$$

Here, $\Phi_i$ is the ionizing EUV photon luminosity from each massive star in units of $10^{49}$ s$^{-1}$ and is dependent on the stellar mass according to the observations of [129,130].

In a simple 'post-processing' analysis of N-body simulations one can follow the mass loss in the discs due to photoevaporation by subtracting mass from the discs according to equations (3.3) and (3.4). Here, we assume each disc is initially 10% of the host star's mass, and subtract mass at every output of the N-body simulation. This allows us to determine the fraction of stars that retain some disc gas mass at each point in time.

In figure 1, we plot the fraction of discs that remain following mass-loss due to photoevaporation in two different star-forming environments. In figure 1a, the initial stellar density is approximately 1000 M$_\odot$ pc$^{-3}$ (thought to be similar to the initial density in the Orion Nebula Cluster), and in figure 1b the initial density is much lower (approx. 10 M$_\odot$ pc$^{-3}$, which is typical of extended OB associations). In both panels, the red lines are discs with initial radii of 10 AU, and the blues lines are discs with initial radii of 100 AU.

Clearly, in very dense environments, we would not expect large (100 AU) discs to retain gas beyond an age of 1 Myr if there were massive stars present [127,131]. This has two very interesting implications for giant planet formation:

  (i) Gas giant planets must form extremely quickly (within 1 Myr), and within 10 AU of the host star where the effects of external photoevaporation will be limited.
 (ii) Alternatively, gas giant planets form exclusively in star-forming regions that do not contain massive stars. This latter point is in direct tension with meteoritic evidence suggesting that our own Solar System was in close proximity to massive stars as planets were forming.

Note that these calculations are likely to be underestimates of the amount of gas lost due to photoevaporation, with the new FRIED grid models showing an even more destructive pattern.

Interestingly, in star-forming regions where massive stars are present, there are very few detections of gas in protoplanetary discs [106,132,133], suggesting that the gas may have already been photoevaporated from these discs.

# 4. Dynamical interactions with passing stars

If a planet has already formed around a star, then in a dense star-forming region (greater than or equal to $100 \, M_\odot \, pc^{-3}$) encounters between passing stars can significantly disrupt the orbits of these planets. Consider a planet, $p_1$, with mass $m_{pl}$ orbiting a star $s_1$ with mass $M_\star$. The planet is on a bound, closed orbit with semimajor axis $a_{pl}$ and eccentricity $e_{pl}$. The orbit has a total energy,

$$|E_{tot}| = \frac{GM_\star m_{pl}}{2a_{pl}}. \tag{4.1}$$

If the planet encounters a second star, $s_2$ with mass $m_{\star,2}$, there are several different possible outcomes of that interaction.

  (i) The planet remains in orbit around the original parent star $s_1$.
 (ii) The planet is stolen by the incoming star, $s_2$, and subsequently orbits this star. Its motion around the secondary star may be prograde or retrograde, and it is likely to have a high eccentricity and inclination.
(iii) The planet is removed from its orbit around the parent star $s_1$, and becomes a free-floating planet within the star-forming region.

(There are variations of these interactions and we refer the reader to table 1 in [134] for a more comprehensive summary.)

In scenario (i), the final orbit of the planet depends to a large extent on the exact energetics of the interaction with the incoming star $s_2$. In any binary system (be it a star–star system, or a star–planet system such as the one considered here), the binding energy of the system $|E_{tot}|$, relative to the Maxwellian energy of the star-forming region, governs the outcome of that interaction—the so-called Heggie–Hills Law [135–137]. The Maxwellian energy is simply the average motion of a star in the region, and can be thought of as being analogous to the average motion of an atom or molecule in a gas in thermal equilibrium. In a star-forming region, this Maxwellian energy can be approximated by

$$E_{Max} = \langle m \rangle \sigma^2, \tag{4.2}$$

where $\langle m \rangle$ is the average stellar mass, and $\sigma$ is the velocity dispersion—the statistical dispersion about the mean velocity—of stars in the region. For a system where the binding energy exceeds the Maxwellian energy, the semimajor axis is likely to decrease, i.e. in our scenario the planet would move closer to its host star. Conversely, when the binding energy of the system is lower than the background Maxwellian energy, the semimajor axis is likely to increase, i.e. in our scenario, the planet would move further away from its host star. Because the binding energy is inversely proportional to the semimajor axis, an increase in the planet's semimajor axis therefore reduces the binding energy and makes the planet more susceptible to further interactions.

However, these three outcomes of an interaction with a passing star do not describe all of the physical processes that occur to planets in dense stellar environments and in order to learn in detail about the effects of stellar interactions on planetary orbits, simulations of the star-forming environment are required.

There are two different approaches to modelling the interactions between planetary systems and passing stars. We will refer to the first approach as a 'direct' calculation, where a brute force approach is adopted. These calculations include the planets in the simulations of the star-forming region, and calculate the gravitational force due to all of the stars in the region on each planet [1,138,139]. This approach is computationally expensive, due to the very different timescales involved. A star will orbit the centre-of-mass of the star-forming region on timescales of Myr, whereas a planetary orbit has timescales of days or years. However, most N-body codes are designed to cope with primordial stellar binary systems, so the planet merely represents a low-mass companion to the star [140–142]. The main disadvantage with this approach is that—aside from several pioneering simulations using graphical processor unit technology—these simulations are limited to one or two planets per star (although there are notable recent exceptions [143,144], which use a hybrid selection of numerical integrators to model four-planet systems around some stars).

The alternative approach is to use Monte Carlo simulations to choose the parameters of a stellar encounter (velocity) with a planetary system, and then follow the planetary system with a numerical integrator to determine the fate of the system [66,145]. These calculations are less computationally intensive, and have the advantage that more than one planet can be modelled around each star.

It is almost impossible to compare these two different approaches. Due to computational constraints, the direct approach cannot be used to model more than one or two planets, while the scattering calculations routinely include four or more planets. Furthermore, the huge parameter space available in these simulations (stellar density, stellar velocity dispersion, initial planetary orbits) means that simulations by different authors are unlikely to overlap so that they are directly comparable. The only thorough comparison between these two techniques for planet disruption calculations is the work in [146]. Even this study uses different initial conditions for the modelled star-forming regions, adopting the Monte Carlo approach for large-N environments (e.g. Globular clusters) and direct N-body for smaller-N environments (e.g. open clusters).

In the Monte Carlo approach, we assume an interaction rate—i.e. how often does a star–planet system encounter another star in the star-forming region—which is written

$$\Gamma = \langle n \rangle \langle \Sigma \rangle \langle \sigma \rangle, \tag{4.3}$$

where $n$ is the stellar number density, $\Sigma$ is the cross section for the interaction and $\sigma$ is the velocity dispersion within the star-forming region. The cross section for the interaction, $\Sigma$, can be written

$$\Sigma = \pi R_{\text{enc}}^2, \tag{4.4}$$

where $R_{\text{enc}}$ is the radius of the encounter, or put more simply, the size of the 'target'—the planet–star orbit. However, the larger orbits (tens of AU) will be more susceptible to interactions that change the planet's orbit, and as such a term that accounts for gravitational focusing is required, such that

$$\Sigma = \pi R_{\text{enc}}^2 \left( 1 + \frac{G(M_\star + M_{\text{int}})}{\sigma^2 R_{\text{enc}}} \right), \tag{4.5}$$

where $M_{\text{int}}$ is the mass of the intruding star.

The earliest work in this area [145] estimated that for a star-forming region with a stellar density of $1000 \, \text{stars} \, \text{pc}^{-3}$, around 10% of planets will experience a disruptive event, although these simulations assumed a very long-lived star-forming region ($10^8$ years, when most star-forming regions have dispersed after $10^7$ years [147]).

This was taken further by examining what the influence of nearby star-forming regions to the Sun would be on our Solar System planets [66]. Their Monte Carlo scattering simulations placed planets of the same mass and semimajor axis as the four giant planets of the Solar System. They computed the cross section for collisions, $\Sigma$, assuming the interactions are typically with an incoming binary star system (as a significant fraction of stars in the Galactic field are in binary systems [10,11,148], and this fraction may be even higher in young star-forming regions [149]). In these simulations, [66] find that Neptune, as the outer planet, is more readily disrupted (despite it having a higher mass than Uranus). Furthermore, they find that disruption of planets is more likely around lower-mass stars, with the cross section for disruption scaling as

$$\langle \Sigma \rangle \sim M_\star^{-1/2}. \tag{4.6}$$

## 4.1. Unbound, or free-floating planets

Despite the unavailability of appropriate comparisons between the direct $N$-body approach and the Monte Carlo scattering experiments, it appears that the scattering experiments are more conservative in their predictions for planetary disruption. This was first noticed in simulations where free-floating planets were created—planets that become unbound from their parent star and then move around the star cluster as a gravitationally distinct entity.

Free-floating planets (FFLOPS) have been observed in star clusters [150–153], including old open clusters such as the Pleiades [154,155], though their physical origin is the subject of much debate [156], with some authors arguing they form from the collapse and fragmentation within the GMC (like stars), whereas other authors argue they form around stars ('like planets') and are then liberated from their host stars.

The latter formation channel was explored in some of the earliest scattering simulations [157,158], and these authors found that the planets were liberated from the host stars at such high velocities that they could not be retained by the gravitational influence of the cluster, which appears at odds with the observations [150]. However, some of the first direct $N$-body simulations [138] predict significant retention of free-floating planets in open clusters, as well as in globular clusters (if planets are able to form in such low-metallicity environments). This was later corroborated by several authors [1,139] who found significant numbers of FFLOPS could be retained by clusters, in contrast to the Monte Carlo approach [159].

Interestingly, the estimates of the number of FFLOPs in the Galactic field can be very high [160], with some authors claiming several free-floating planets per star in the Milky Way Galaxy (though see [161,162]). Such a population cannot be created entirely by instabilities within planetary systems leading to the ejection of planets (so-called planet–planet scattering) [163], so an additional mechanism would be required. Star-forming regions with stellar densities of the order 1000 stars pc$^{-3}$ can liberate up to 10% of planets [1,139], although this is a lower limit because these simulations typically only contain one planet around each star and an interaction that disrupts a planet so that it becomes free-floating is likely to have a significant impact on the entire planetary system [144,164].

## 4.2. Disruption of orbits

If an interaction with a passing star is not strong enough to liberate the planet completely from its orbit and create an FFLOP, the orbit may be significantly altered. Early simulations predicted that to disrupt the orbits of planets, the star-forming region must be either very dense (greater than 1000 M$_\odot$ pc$^{-3}$) [158], or that the host star must remain in a star-forming region for a significant amount of time (hundreds of Myr, i.e. greater than $10^8$ years). As we have discussed, these early simulations generally adopt the Monte Carlo scattering approach, which tends to be conservative in its predictions for disrupting orbits, and these simulations assumed a very simplistic set of parameters for the cluster environment.

Typically, these early simulations assumed a uniform velocity dispersion across the cluster of approximately 1 km s$^{-1}$, as well as a smooth, centrally concentrated density profile for the stars in the cluster (usually a Plummer [165] or a King [166] profile, although some authors [96] use variations of these types of distributions). It must be emphasized that these smooth distributions tend to be excellent models for star clusters as we observe them at older ages, but fail to capture the dynamics of stars in young star-forming regions.

We know from both observations and simulations [37,167–169] that young star-forming regions possess significant degrees of spatial and kinematic substructure. Visually, they appear to have a very clumpy, or ragged distribution and the clumps of stars tend to have very correlated velocities on local scales [170], but can have very different velocities to (clumps of) stars in other areas of the star-forming region. Dynamical relaxation erases this substructure on a timescale that scales with the median local density of a star-forming region [35,42,43,171,172], and if the global virial ratio of the stars is low, the star-forming region may fall in on itself to form a smooth, spherical cluster [43,173]. The point is that if one assumes a smooth distribution for disruption of planetary orbits, much of the physics that occurs in star-forming regions as planets are forming is ignored. Loosely speaking, this means that spatially and kinematically substructured star-forming regions are more dynamically active, and planetary orbits can therefore be affected at much lower stellar densities than previously thought (though still upwards of 100 M$_\odot$ pc$^{-3}$).

In figure 2, we show the typical outcome of a simulation with a stellar density initially in the region of 1000 M$_\odot$ pc$^{-3}$, where each low-mass (less than 3 M$_\odot$) star has been assigned a planet initially on a zero eccentricity orbit with a semimajor axis of 30 AU [1]. Following 10 Myr of dynamical evolution in

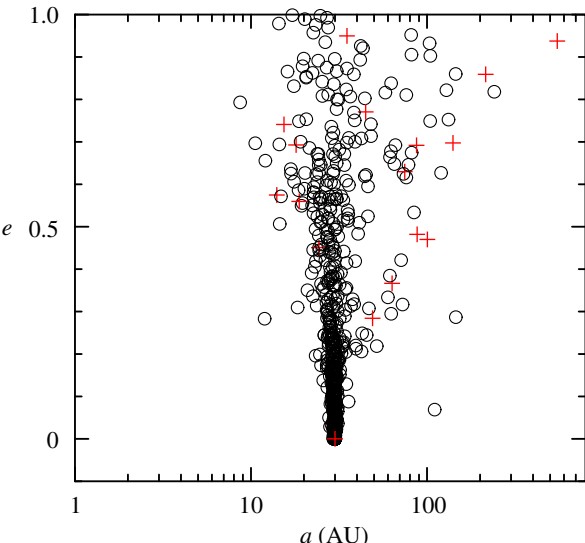

**Figure 2.** Disruption of planetary orbits in a dense (approx. $1000 \, M_\odot \, pc^{-3}$) star-forming region. The planets are 1 $M_{Jup}$, and placed on initially circular orbits ($e = 0$) at 30 AU from their parent star. The panel shows the distribution of eccentricity $e$ and semimajor axis $a$ of the planets following 10 Myr of dynamical evolution in the star-forming region. The open circles indicate planets that are still orbiting their parent stars. In this particular simulation, 10% of the planets become free-floating, i.e. they are no longer gravitationally bound to a star. A handful of these planets are then re-captured around other stars. Examples of these captured planets are shown by the red plus symbols in the panel. Adapted from Parker & Quanz 2012 [1].

the star-forming region, during which time the substructure is erased due to violent relaxation and the region collapses to form a cluster, and then expands, the orbits of up to 20% of the planets are significantly disrupted. Primarily, the eccentricity of a planet can be raised from zero to anywhere between zero and unity, and a smaller number of systems (10%) have their semimajor axes changed by $\pm 10\%$. Several groups of researchers find similar results when following the evolution of planets in substructured star-forming regions [139,174].

## 4.3. Secondary effects

In addition to direct interactions between a passing star and a planetary system, planets can be disrupted by so-called secondary dynamical effects, often if they orbit a star that is part of a binary system. In this scenario, the passing star changes the orbit of the binary, which could lead to future dynamical instabilities in the planetary system.

Typically, a planet orbiting a component within a binary system is stable if its semimajor axis is less than one-third of the semimajor axis of the binary [63], although this depends on the exact orbital configuration of the binary. However, if an interaction either increases the eccentricity of the binary, or hardens the binary (decreases the semimajor axis) then the planet's orbit may no longer be dynamically stable.

One of the most notable secondary effects is the Kozai–Lidov mechanism, or the von Zeipel–Lidov–Kozai (vZLK) mechanism [175–179]. The vZLK mechanism was first developed to explain peculiarities in the orbits of asteroids, but is applicable to different types of three-body systems, including planets within binary stars. The vZLK mechanism posits that if the inclination of two stars in a binary increases by more than 39.23°, then a transfer of angular momentum from the secondary star in the binary onto the planet occurs. This, in turn, causes the eccentricity and inclination of the planet to oscillate wildly, creating dynamical instabilities in the planetary system which can lead to orbit crossings and ejection of the planets entirely from the system.

If the Solar System underwent oscillations due to the vZLK mechanism, then it is probable that both Uranus and Neptune would be ejected, with Saturn and Jupiter pumped to higher eccentricities [180]. Given a typical population of binary stars in a dense star cluster, it is likely that up to 10% of binary stars could undergo Kozai cycles [181], which usually occur on timescales less than 1 Myr [182]. Planets in binary systems occur as frequently as planets orbiting single stars [64], so the vZLK mechanism and other secondary effects could be important.

## 4.4. Other effects

The most common avenue of planet formation is thought to be the core accretion scenario, where bodies grow by dust coagulation [183] and pebble accretion [184]. However, planets could also form from the fragmentation of circumstellar discs [185]. If planets do form via fragmentation, then if interactions with passing stars occur during the protoplanetary disc stage then they can suppress planet formation [186]. Furthermore, some authors find that in discs affected by stellar flybys, the planets that form are more massive and at larger orbital distances [187].

Interactions with passing stars can both induce planet–planet scattering [188], as well as regulate its effects. Some planet–planet scattering events coupled with fly-bys from passing stars in relatively low-density environments find that the passing stars stabilize the orbits of planets ejected from the scattering process, thus creating 'Oort planets' [189].

While star clusters and star-forming regions, in general, are destructive to planetary systems, it is worth noting that only between 10 and 20% of systems are affected by direct interactions. Even assuming secondary effects such as the Kozai–Lidov mechanism can further destabilize planetary systems, more than 50% of planetary systems in dense stellar environments would remain unscathed. Over the typical lifetimes of star clusters, Solar Systems can be disrupted [164], but many planets survive in these environments [190,191], so it is unsurprising that planets are observed in open clusters [192,193].

## 5. The birth environment of the Solar System

While we are primarily interested in a discussion of where all planet-hosting stars may have formed, this is naturally framed around the anthropic question of whether our Sun and its planets formed in a fashion typical of many other stars in the Universe, or whether we are somehow special in the context of all other planetary systems. In the following, we will assume that the Sun formed from a molecular cloud that collapsed and fragmented, but we will make no *a priori* assumptions about the initial stellar density, or total mass of the cloud.

The main planets in our Solar System (Mercury, Venus, Earth, Mars, Jupiter, Saturn, Uranus and Neptune) all have prograde orbits, very low orbital inclinations and very low orbital eccentricities. Furthermore, unlike many of the observed exoplanetary systems, there are no massive planets on very close orbits, implying that planet–planet scattering, and/or significant orbital migration, have not taken place in the Solar System. (Note that a small degree of orbital migration may have taken place, and has been invoked to explain certain features of the Solar System [194,195].)

However, the Solar System appears not to have been completely unscathed by its birth environment. The remainder of the Sun's circumstellar disc, the Edgeworth–Kuiper Belt, has a rather abrupt edge at 50 AU [196,197], where we would expect a more gradual drop-off in surface density of the disc. Furthermore, some Kuiper Belt objects, such as Sedna, have highly inclined orbits out of the plane of the Solar System, suggestive of a reasonably close encounter ($400 < a_{enc} < 1000$ AU) with a passing star [198], although [199] argue the encounter could have been as close as 50–150 AU.

More recently, some authors [200] have suggested that the clustering in argument of perihelia of some Edgeworth–Kuiper Belt objects (EKBOs) suggests dynamical stirring by a much more massive ($20\,M_\oplus$) planet (the so-called Planet 9, see [201] for a review). Whilst some authors suggest the unseen planet has been able to form at such large distances from the Sun [202], in general, massive planet formation is thought to occur closer in. Several authors have proposed that Planet 9 was either captured or stolen from another star [203,204], which would happen when the Sun was still embedded in its natal star-forming region. For this to happen, the stellar densities must be reasonably high ($100\,M_\odot\ pc^{-3}$), but not so high that the inner planets would be disrupted.

An interesting experiment is to determine the fate of the Solar System if it were in a much more dense environment ($1000\,M_\odot\ pc^{-3}$). If we place simulated Solar Systems in star-forming regions of varying stellar density, we obtain several interesting results [144]. First, Neptune and Uranus are more susceptible to liberation from the Solar System than the inner planets.

Secondly, the presence of Jupiter has several interesting implications for the habitability of Earth-like terrestrial planets. In a fairly low stellar density, Jupiter acts as a shield from perturbations from passing stars. However, when the stellar density is much higher, Jupiter itself is perturbed to such an extent that it negatively impacts the habitability of Earth.

The architecture of the Solar System therefore argues for a fairly benign, or low density stellar environment for the formation of the Sun. However, there are some tantalizing clues to the birth environment of the Sun present in some of the oldest objects in our Solar System, the chondritic meteorites [205–207]. These objects date from the epoch of planet formation around the Sun, and contain enhanced levels of short-lived radioisotopes (SLRs) relative to the ISM. Two SLRs, $^{26}$Al and $^{60}$Fe are particularly interesting, as they have very short radioactive half-lives (0.7 Myr and 2.6 Myr, respectively [208]). The presence of SLRs with such short half-lives means that this $^{26}$Al- and $^{60}$Fe-rich material must have been incorporated into the Solar System either as it was forming, or just after planets started to form.

$^{26}$Al is produced by cosmic ray spallation [209]—high energy cosmic rays from the Sun that induce non-thermal nuclear fission—but if this were the dominant production channel of $^{26}$Al we would not expect it to be homogeneously distributed throughout the Solar System, which it appears to be. This SLR can also be produced during the asymptotic giant branch (AGB) phase—an advanced stellar evolutionary phase of Sun-like stars. However, AGB stars are not found in star-forming regions, nor was the Sun ever likely to have had a chance encounter with one [210]. Both of these processes cannot produce $^{60}$Fe in the abundances observed in the Solar System.

$^{26}$Al and $^{60}$Fe are produced in the cores of very massive stars (greater than $20\,M_\odot$, [211]), and are released either during the supernova explosion of these stars at the end of their lives (typically 8–9 Myr or less, with more massive stars exploding earlier [212]), or in the case of $^{26}$Al, emitted via the stellar wind during the massive star's evolution off the Main Sequence.

There are two different mechanisms for seeding the Solar System with these SLRs. The first is via the capture of the $^{26}$Al and $^{60}$Fe material by the Sun's protoplanetary disc after the massive star(s) explodes as a supernova, or via the capture of $^{26}$Al from the stellar wind [213,214]. In this 'disc enrichment' scenario, the Sun is required to be 'in the right place at the right time', i.e. close enough to the massive star to collect enough of the material without being destroyed by the supernova blast wave itself [215,216]. This scenario also requires the Sun to have been born in a star-forming region with at least one massive star. Typically, this would mean the Sun formed with hundreds, if not thousands, of other stars, as massive stars only occasionally form in low-mass star-forming regions (although there are some observed examples).

The second mechanism posits that the Sun formed in a giant molecular cloud that was already pre-enriched in $^{26}$Al and $^{60}$Fe [217,218]. In order for this to happen, the Sun must have been a member of an inter-generational star-forming sequence. First, a population of stars forms in the GMC. The most massive of these explode, triggering a second burst of star formation in which the stars that form are enriched in $^{60}$Fe. The second generation must contain at least one massive star (greater than $20\,M_\odot$), which undergoes a Wolf–Rayet (WR) phase [219], during which its stellar wind is rich in $^{26}$Al. This WR star, in turn, triggers the formation of a third generation of stars, one of which is our Sun. This mechanism ensures the levels of $^{26}$Al and $^{60}$Fe observed in the Solar System are reproduced, and is based on the notion that star-forming regions regularly trigger additional star formation [220].

Both the disc enrichment, and sequential enrichment scenarios have their drawbacks. The disc enrichment model requires a very fine-tuned set of circumstances; the Sun must have been in the right place at the right time, the disc cannot be destroyed by the supernova. Even the most massive stars explode as supernovae after approximately 4 Myr, by which point the protoplanetary disc may have been significantly depleted [88,89]. In the sequential enrichment scenario, the giant molecular cloud must remain for around 15 Myr, which is unlikely in the presence of feedback from the most massive stars [221,222]. Furthermore, multiple generations of stars would be observed to have significant age spreads, or dichotomies in ages [223], which are not observed in star-forming regions [224,225].

However, based on our consideration of disc photoevaporation, the biggest drawback with both enrichment scenarios is that massive stars will be close (less than 1 pc) from the young Sun, and FUV radiation will potentially destroy the gas component of the protosolar disc before enrichment can take place. There is, therefore, a significant tension here; on the one hand, enrichment from massive stars is required to apparently drive the internal evolution of Earth [226], but photoionizing radiation is likely to significantly disrupt the formation of Jupiter and Saturn. Resolving this tension remains an active topic of research in the field [2,127,131,227], with the solution being that perhaps the Sun's disc was truncated to a radius that allowed the formation of Jupiter and Saturn, but severely depleting the outer regions of the disc.

We know from our earlier discussion that most star-forming regions disperse into the Galactic field within the first 10 Myr, and so at face value it would appear nearly impossible to pinpoint the Sun's birth environment, as our star's stellar siblings have probably migrated large distances within the Galactic disc [228].

However, work has demonstrated that stars that originate from the same stellar nursery as the Sun could in principle be traced using their unique chemical signatures [229–231]. This method, known as 'chemical tagging' [232], is largely in its infancy [233], but high-resolution spectral surveys mean that data of the required quality and quantity are now readily available [234–237].

Indeed, some authors have claimed one of the Messier objects, the M67 open cluster, may be the birth cluster of the Sun, based on its similar age and almost identical chemistry of the stars [238]. We know from observations that planets are present around stars in this cluster including around Solar twins [239].

As M67 is a relatively massive open cluster, it is likely that the young Sun may have experienced significant radiation fields that could have evaporated the gas in its protoplanetary disc. However, this also suggests that enrichment in $^{26}$Al and $^{60}$Fe could have occurred. If the Sun did originate in M67, it may have been ejected early on, so that subsequent interactions in the cluster did not disrupt the outer planets [240]. Interestingly, attempts have been made to calculate when and where in time the Sun and M67 intersect, without much success [241], arguing that perhaps the Sun may have originated elsewhere (and the similarity in the chemistry of M67's stars to the Sun could be chance [233]). Alternatively, a collision with a GMC could have disrupted the orbit of M67 itself, making it impossible to rule out the Sun's origin in this cluster [242].

# 6. Conclusion and outlook

While there are other factors that determine the amount of outside influence a planetary system can expect during formation, the main variable is the stellar density in the star-forming region in question. The stellar density determines the number of encounters a star and its planetary system can expect to experience, and also governs the flux of EUV and FUV radiation that can potentially photoevaporate protoplanetary discs.

Studies of star-forming regions suggest that most have initial stellar densities of at least $100\,M_\odot\,\mathrm{pc}^{-3}$ (a factor of 1000 higher than the stellar density in the Sun's current environment). However, there are some star-forming regions that have lower densities, as well as a small fraction of star formation that occurs in very dense (greater than $1000\,M_\odot\,\mathrm{pc}^{-3}$) regions.

If massive stars are present, then we expect planet formation to be disrupted or altered by photoevaporation in star-forming regions of all densities. In low mass, low density star-forming regions, we do not expect many detrimental effects on the planet formation process, but crucially, we do not really know what fraction of planet hosting stars form(ed) in such benign environments.

In moderate to high density star-forming regions ($100\,M_\odot\,\mathrm{pc}^{-3}$), interactions between stars are common enough that the orbits of newly formed planets can be altered, either directly by a perturbation from a passing star, or indirectly through secondary dynamical processes, such as the vZKL mechanism. It is only in star-forming regions with extremely high densities greater than $1000\,M_\odot\,\mathrm{pc}^{-3}$ that we expect significant truncation of protoplanetary discs due to encounters with passing stars. Typically, star-forming regions with such high densities are also very massive and therefore likely to contain massive stars, in which case we would expect the discs to be severely affected by photoevaporation.

The next steps in this research field require a multi-faceted approach. Ideally, one would like to follow the formation of planets from a disc within a global hydrodynamical simulation of star formation in a GMC. Unfortunately, this is some way off, but promising steps are being made to model the interaction of stars in an $N$-body simulation with an evolving background gas potential [243,244]. This will enable us to model the formation and evolution of star-forming regions from the early gas-dominated phase to the later gravitationally dominated phase (i.e. what we traditionally think of as being a star cluster). Simultaneously, the resolution of hydrodynamical simulations can also be used to model the formation and evolution of protoplanetary discs within a star cluster [245]. In a similar spirit, $N$-body simulations can now incorporate the integrations of multi-planetary systems, thanks to the use of different numerical integration schemes within the same simulation [143,246].

At the same time, the advent of ALMA has revolutionized observational studies of protoplanetary discs [68], and the discovery and characterization of exoplanetary systems continues apace. The next generation of ground- and space-based telescopes are likely to further our understanding of the Solar System in the context of other planetary systems. Perhaps (and I hope) we will reach the stage where we can disentangle the internal processes that alter a planetary system (migration, internal photoevaporation from the host star, planet–planet scattering) from external influences, and search for these differences in observations.

With so many promising avenues for future research, I very much hope (and expect!) that this review will be obsolete in a very short space of time.

Data accessibility. The data used to produce the figures is available on request to the author. No new data, code or other digital materials were generated in the preparation of this work. For the recreation of figures, the original data can be found in Nicholson et al. [127] (figure 1) and Parker & Quanz [1] (figure 2) and these raw data are available on request to the author.

Competing interests. I declare I have no competing interests.

Funding. R.J.P. is supported by a Royal Society Dorothy Hodgkin research fellowship.

Acknowledgements. I wish to thank the amazing staff at Malin Bridge Primary School, Sheffield, whose hard work enabled my son to return to classes during the UK's COVID-19 lockdown, giving me the extra time to complete this review. Thanks to Emma Daffern-Powell, Christina Schoettler, Rhana Nicholson and Tim Lichtenberg for inspiring me to write this article. Thank you to all of the undergraduate students I have lectured on these topics, and those that have done their projects under my supervision. I am grateful to the two anonymous referees for their comments and suggestions on the original manuscript. And finally, thanks to Sascha Quanz for his throwaway question that led to me working in this field.

Disclaimer. Views and opinions expressed in this article are solely the author's.

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
