## [Reviewer comments · Royal Society Open Science]

Review History

RSOS-201271.R0 (Original submission)

Review form: Reviewer 1

Is the manuscript scientifically sound in its present form?

Yes

Are the interpretations and conclusions justified by the results?

Yes

Is the language acceptable?

Yes

Do you have any ethical concerns with this paper?

No

Have you any concerns about statistical analyses in this paper?

No

Recommendation?

Accept with minor revision (please list in comments)

Comments to the Author(s)

This paper provides an updated review of how cluster environments can influence the planetary systems forming within them. The review is comprehensive and clearly written, and will provide

a useful addition to the literature (especially for researchers just entering the field, including students). The paper is in good shape, and can be accepted for publication after the following relatively minor considerations have been taken into account. Although the following list is somewhat long and somewhat wordy, hopefully these issues will be straightforward to address:

The review focuses on dynamical interactions and effects due to the radiation fields provided by the cluster environment. In addition to these effects, clusters can also provide 'particle' background effects. Since the paper cannot cover everything, this omission is OK, but it should still be briefly mentioned in defining the scope of the review. The review does mention the example of short-lived radioactive nuclei in conjunction with the birth environment of the solar system (section 5) but not for the general case. In addition to their signatures in meteorites, which is rather specific to our solar system, these nuclei provide a source of ionization, which affects MRI and chemistry during the planet formation process. In addition, cluster environments can both enhance the cosmic ray flux (by trapping particles from supernovae) and shield the cosmic rays (via magnetic fields). Since the cosmic rays provide an important source of ionization, they are important for star formation (e.g., coupling the magnetic field during collapse) and planet formation (see above).

Page 3, line 33: GMCs are not really collapsing as a whole, at least not in an organized monolithic way (this finding dates back to Zuckerman and Palmer 1974). The GMCs are dynamically evolving, but 'collapsing' is not really the right description.

Page 3, line 34: On a similar note, while the Jeans mass does provide a scale, saying that the cloud fragments into one solar mass pieces is overly simplistic. To see this, use the stated temperature (tens of K) and a typical density of the molecular cloud (say 1000 particles per cc) and calculate a Jeans mass. You will find that it's larger than a typical star. You can always choose a larger density to get the answer that you want, but then you have to justify why you are [1] using a density that is larger than the typical density of the cloud, and [2] explain why you are using only a single density, instead of the rather wide range of densities sampled by the cloud. Notice also that the Jeans mass does not depend on things like angular momentum and magnetic fields, ingredients that we know are important for the star formation process. So, while Sir James Jeans is certainly correct, invoking the Jeans mass to explain stellar masses is probably more misleading than enlightening.

Page 4, first full paragraph: In the discussion of stellar densities, it would be useful to point out that clusters have a density distribution, so that forming solar systems will experience a range of stellar densities. If the cluster lives long enough, then the members would sample the full distribution of stellar densities. But disks last only a few to several Myr, and planets form on the same time scale, and dynamical relaxations time are longer, so not all solar systems will fully sample the available densities in the cluster.

Page 4: On a related note, while the stellar density is important for determining the degree of disruption from both dynamical interactions and radiation, the time spent in the high density environment is also important. For example, the second full paragraph (that explains the effects of different stellar densities) must be assuming something about the time spent in that environment. It's important to be more quantitative. If the time is only 1 Myr, which is on the low end for both disk lifetimes and planet formation, then the effects will be much less important than if the time is 10 Myr or 100 Myr.

Page 4: Paragraph at lines 45 to 52: This paragraph asserts that the Sun formed at redshift $z=1$ when star formation (on a cosmic scale) was more active. Perhaps I am on the wrong page, but I find that the redshift for the formation of the Sun is more like $z=0.43$ (although the result depends on what cosmological parameters are assumed). Perhaps the author is confusing the age of the universe and the look back time? The age of the universe was about 4 Gyr at redshift $z=1$, but that was 9 Gyr ago, about twice the age of the solar system. In any case, the cosmology used to get

this result should be specified. And if $z=0.43$, as I suspect, then the rest of the paragraph is less relevant.

Page 5: Equation (2.2) seems to have an error/typo: Maybe the cluster mass should be on top? Even in that case, if one uses a cluster mass $M_c = 2000 M_{\text{sun}}$ (like the ONC) and a galaxy mass $M_g = 10^{10} M_{\text{sun}}$, and $R_G = 8 \text{ kpc}$ (like the solar circle), then the Jacobi radius $r_J = 47 \text{ pc}$. The radius of the ONC is usually quoted as $r = 2 \text{ pc}$ (e.g., Hillenbrand and Hartmann 1997). Which leads to the question: How does an outer boundary at 47 pc (from external galactic tides) affect a star cluster with a radius of only 2 pc?

Page 6, line 37: Although the review focuses on dynamical interactions and evaporation from the cluster radiation, the disks also lose mass by disk accretion onto the star, additional evaporation by stellar photons, and even by locking up some mass into their planets. For completeness, these channels should be mentioned, and this point in the text (right before section 3a) would be a good place.

Page 6, line 52: The text says that stars can undergo multiple interactions. It would be useful to add some further explanation to clarify the conditions required to get multiple interactions. The interaction rate is (approximately) $\Gamma = n \sigma v$ where $n = 100 \text{ pc}^{-3}$ is the stellar density, $v = 1 \text{ km/s}$ is the typical interaction speed, and the cross section in this example is about $\sigma = \pi (300 \text{ AU})^2$ (from line 44). The text should thus discuss how much time it takes to get one interaction close enough to truncate the disk, and then how likely it would be to get multiple interactions.

Page 7, line 20: The text states that 200 AU disks are common. Some discussion of the distribution of observed disk sizes, along with references, should be given here. Many papers quote 100 AU as the typical disk size and find that 200 AU is more of an upper limit. The difference matters, because if most disks start out with radii of 100 AU (like HL Tau, which is discussed earlier) then they do not need to be truncated to reach 100 AU sizes...

Page 9, lines 34 to 40: The text lists two possible implications for planet formation, either planets form quickly, or they form in regions with low radiation levels. But, based on the numbers presented in the text, isn't there a third possibility? The evaporation process becomes inefficient and does not operate effectively for radial locations $r < 10 \text{ AU}$ (page 8, line 53). So why can't planets just form at radial locations in disks where $r < 10 \text{ AU}$? Planet formation models often prefer to have planets form just outside the ice-line for a number of reasons, so the best place to form planets is at about 5 AU (where evaporation is not so effective).

Page 12: On freeing floating planets:

Planet ejected from their solar systems will have a distribution of ejection speeds, where the distribution can be approximated by the form $dP/du = 4u/(1+u^2)^3$ (e.g., Moorhead and Adams 2005). As a result, if enough planets are ejected, then some fraction of them will be retained in the cluster as freely floating planets.

On the other hand, the paper by Sumi et al. (2001) (ref. [155]) has been updated by the microlensing collaboration and the estimates for the number of freely floating planets has been revised downward.

Page 13: Secondary effects: The quoted stability criterion of 'one third the binary semimajor axis' is not stringent enough. First, the paper by Holman and Weigert (Ref. [59]) only integrates over 10^4 binary orbits, which is not nearly long enough to explore long term stability, and it also corresponds to a different physical time scale for each type of binary. A better criterion is that the orbit of the planet must be less than some fraction of the binary periastron. In order to make an earth-like orbit stable for 4.6 Gyr, the age of the solar system, the binary periastron must be about 7 AU or greater (for earths orbiting the primary). So interactions that increase binary eccentricity

will lead to disruption of planetary orbits. Finally, you can easily tell that 'one third of the binary semimajor axis' is not enough: If the binary has $e = 2/3$, then it can cross the planetary orbit. (And notice also that the binary eccentricity distribution $dP/de = 2e$ -- so that high- e binaries are in fact likely).

Page 15, line 30: It seems to be an oversight so discuss Planet Nine, but not cite any of the (many!) papers by Batygin and Brown. Maybe cite their recent review article (Batygin et al. 2019, Physics Report) and refer the reader to the references therein.

A general comment on the birth environment of the solar system: Many papers (like the Nice model) indicate that the giant planets formed within about 20 AU of the Sun. Since photoevaporation is inefficient that close to the Sun, isn't there a scenario where the birth cluster evaporates the solar nebula down to 25 or 30 AU, while the planets are forming at smaller radii? Note that Uranus and Neptune don't really contain any gas, so the solar nebula really just needs to retain gas at 5 to 10 AU for Jupiter and Saturn. Moreover, such a truncated solar nebula would help explain the deficit of material in the Kuiper Belt. The current text reads as if disks are either evaporated, or not, but does not leave room for partially evaporated (and thus partially truncated) disks. I have always thought that this in-between scenario was the most likely.

Page 16, Section 6, paragraph 1: While density is certainly *one* of the most important variables, I would think that the time spent in the high density environment would be on an equal footing.

Review form: Reviewer 2

Is the manuscript scientifically sound in its present form?

Yes

Are the interpretations and conclusions justified by the results?

Yes

Is the language acceptable?

Yes

Do you have any ethical concerns with this paper?

No

Have you any concerns about statistical analyses in this paper?

No

Recommendation?

Accept with minor revision (please list in comments)

Comments to the Author(s)

Referee report for

„The birth environment of planetary systems“

by Richard J. Parker

This review article summaries very well the state-of-the-art of the research on the influence of the environment on young (forming) planetary systems. Th author presents this topic in a very structured approach that facilitates obtaining an overview of this subject. Therefore I recommend publication. However, I would suggest for the authors to consider the following points:

Introduction: Here one has the impression that the author is giving a talk to an audience of students. The style he choose might work in such a situation, however, I think the many colloquial expressions and personal reminiscences of his own thinking as student do not work out for a review article. I suggest to rephrase some of the sentences.

Throughout the author uses the term „photo-evaporation“ in the meaning of „external photo-evaporation“. For the expert this is clear, however, as this review is aimed to help newcomers in the field, it should be pointed out early on in text that photo-evaporation by the star itself and external photo-evaporation can lead to disc destruction, but that throughout the text only external photo-evaporation is considered.

Star formation: Here I find that there are too few references to the general processes of the formation of clusters/associations and too many on issues that are of minor importance in a review article. For example, it would be useful to a novice to the field to have a reference to the typical masses and sizes of fragments (page 4, line 33), whereas he/she would be overwhelmed with 12 references to the relation between cluster mass and the mass of the most massive star, which features not very prominent in the rest of the paper.

As the authors points out the stellar density is the key parameter that determines to which degree the environment influences (forming) planetary systems. This means that differences between long- (> 50 Myr) and short- (<10 Myr)-lived stellar groups exist and that the formation, expansion of stellar groups and time scales on which this happens are essential for determining the influence of the environment. There exist different theories concerning all these points. The author tries to smooth over the various conflicts, without pointing them out. I think that newcomers should be made aware of the openness of these issues, rather than leaving them unaware of them.

Photo-evaporation: There are several groups working on the effect of external photo-evaporation. I think there is a too strong emphasis on the FRIED simulation. This should be supplemented by mentioning other works.

Eq. 3.3 and 3.4 were used nearly 20 years ago to describe the effect of external photo-evaporation, the authors themselves pointed out than that using this equations leads to overestimating the effect. Therefore, it is probably not really helpful to present them as state-of-the-art to today's students.

Minor points:

page 1 line 22. The sentence „Recent observations form ALMA suggest that planet formation may already be well under way after only 1 Myr of a star's life“ gives the, perhaps wrong, impression that all planets form within just 1 Myr. This is clearly overstating current knowledge. I think that in some cases planets might form as early as 1 Myr after a star formed reflects more the situation.

page 5 line 12: „lower-mass stars like the Sun“ is misleading as the mass of the Sun is about twice as high as the average stellar mass in a stellar group.

page 5 paragraph 3: it is not always clear whether it is referred to the local or the average density in the stellar group and references to the density are not given. same applies for paragraph 5 on the same page.

page 7 paragraph 3 and 4: the concept of the MMSN is here not very helpful because many (if not most) stars are surrounded by discs with masses below the MMSN. There are many different theories trying to explain that. Either one goes into that discussion or leaves the MMSN out altogether.

Page 7 paragraph 6: The early results were only for equal-mass stars, unfortunately, they were later on used for other mass ratios. One should avoid that this misunderstanding continues.

page 9 paragraph 5: The dependence on the distance to the star is misleading without mentioning that it happens only on the side of the disc that is nearest to the massive star. Thus it is an asymmetric process.

page 10 argument (ii): this argument does not catch here, because one cannot generalise necessarily from our solar system to all other planetary systems

page 17 paragraph 7: The argument that the Sun formed in a short-lived stellar group (< 10 Myr) is weak as most long-lived stellar groups disperse on time scales of a few 100 Myr which is short in comparison to the Sun's age of 4.6 Gyr.

There are some incomplete sentences in the manuscript with words missing. This requires urgently changing before publication.

Decision letter (RSOS-201271.R0)

Dear Dr Parker,

On behalf of the Editors, we are pleased to inform you that your Manuscript RSOS-201271 "The birth environment of planetary systems" has been accepted for publication in Royal Society Open Science subject to minor revision in accordance with the referees' reports. Please find the referees' comments along with any feedback from the Editors below my signature.

Please submit your revised manuscript and required files (see below) no later than 7 days from today's (ie 11-Aug-2020) date. Note: the ScholarOne system will 'lock' if submission of the revision is attempted 7 or more days after the deadline. If you do not think you will be able to meet this deadline please contact the editorial office immediately.

on behalf of Dr Mark Wyatt (Associate Editor) and Professor Rob Ivison (Subject Editor)
openscience@royalsociety.org

Associate Editor Comments to Author (Dr Mark Wyatt):

Both reviewers are positive, but make several comments that will improve the accuracy and clarity of the paper, and which should be taken into account when submitting a revised version.

Reviewer comments to Author:

Reviewer: 1

Comments to the Author(s)

This paper provides an updated review of how cluster environments can influence the planetary systems forming within them. The review is comprehensive and clearly written, and will provide a useful addition to the literature (especially for researchers just entering the field, including students). The paper is in good shape, and can be accepted for publication after the following relatively minor considerations have been taken into account. Although the following list is somewhat long and somewhat wordy, hopefully these issues will be straightforward to address:

The review focuses on dynamical interactions and effects due to the radiation fields provided by the cluster environment. In addition to these effects, clusters can also provide 'particle' background effects. Since the paper cannot cover everything, this omission is OK, but it should still be briefly mentioned in defining the scope of the review. The review does mention the example of short-lived radioactive nuclei in conjunction with the birth environment of the solar system (section 5) but not for the general case. In addition to their signatures in meteorites, which is rather specific to our solar system, these nuclei provide a source of ionization, which affects MRI and chemistry during the planet formation process. In addition, cluster environments can both enhance the cosmic ray flux (by trapping particles from supernovae) and shield the cosmic rays (via magnetic fields). Since the cosmic rays provide an important source of ionization, they are important for star formation (e.g., coupling the magnetic field during collapse) and planet formation (see above).

Page 3, line 33: GMCs are not really collapsing as a whole, at least not in an organized monolithic way (this finding dates back to Zuckerman and Palmer 1974). The GMCs are dynamically evolving, but 'collapsing' is not really the right description.

Page 3, line 34: On a similar note, while the Jeans mass does provide a scale, saying that the cloud fragments into one solar mass pieces is overly simplistic. To see this, use the stated temperature (tens of K) and a typical density of the molecular cloud (say 1000 particles per cc) and calculate a Jeans mass. You will find that it's larger than a typical star. You can always choose a larger density to get the answer that you want, but then you have to justify why you are [1] using a density that is larger than the typical density of the cloud, and [2] explain why you are using only a single density, instead of the rather wide range of densities sampled by the cloud. Notice also that the Jeans mass does not depend on things like angular momentum and magnetic fields, ingredients that we know are important for the star formation process. So, while Sir James Jeans is certainly correct, invoking the Jeans mass to explain stellar masses is probably more misleading than enlightening.

Page 4, first full paragraph: In the discussion of stellar densities, it would be useful to point out that clusters have a density distribution, so that forming solar systems will experience a range of stellar densities. If the cluster lives long enough, then the members would sample the full distribution of stellar densities. But disks last only a few to several Myr, and planets form on the same time scale, and dynamical relaxations time are longer, so not all solar systems will fully sample the available densities in the cluster.

Page 4: On a related note, while the stellar density is important for determining the degree of disruption from both dynamical interactions and radiation, the time spent in the high density environment is also important. For example, the second full paragraph (that explains the effects of different stellar densities) must be assuming something about the time spent in that

environment. It's important to be more quantitative. If the time is only 1 Myr, which is on the low end for both disk lifetimes and planet formation, then the effects will be much less important than if the time is 10 Myr or 100 Myr.

Page 4: Paragraph at lines 45 to 52: This paragraph asserts that the Sun formed at redshift $z=1$ when star formation (on a cosmic scale) was more active. Perhaps I am on the wrong page, but I find that the redshift for the formation of the Sun is more like $z=0.43$ (although the result depends on what cosmological parameters are assumed). Perhaps the author is confusing the age of the universe and the look back time? The age of the universe was about 4 Gyr at redshift $z=1$, but that was 9 Gyr ago, about twice the age of the solar system. In any case, the cosmology used to get this result should be specified. And if $z=0.43$, as I suspect, then the rest of the paragraph is less relevant.

Page 5: Equation (2.2) seems to have an error/typo: Maybe the cluster mass should be on top? Even in that case, if one uses a cluster mass $M_c = 2000 M_{\text{sun}}$ (like the ONC) and a galaxy mass $M_g = 10^{10} M_{\text{sun}}$, and $R_G = 8 \text{ kpc}$ (like the solar circle), then the Jacobi radius $r_J = 47 \text{ pc}$. The radius of the ONC is usually quoted as $r = 2 \text{ pc}$ (e.g., Hillenbrand and Hartmann 1997). Which leads to the question: How does an outer boundary at 47 pc (from external galactic tides) affect a star cluster with a radius of only 2 pc?

Page 6, line 37: Although the review focuses on dynamical interactions and evaporation from the cluster radiation, the disks also lose mass by disk accretion onto the star, additional evaporation by stellar photons, and even by locking up some mass into their planets. For completeness, these channels should be mentioned, and this point in the text (right before section 3a) would be a good place.

Page 6, line 52: The text says that stars can undergo multiple interactions. It would be useful to add some further explanation to clarify the conditions required to get multiple interactions. The interaction rate is (approximately) $\Gamma = n \sigma v$ where $n = 100 \text{ pc}^{-3}$ is the stellar density, $v = 1 \text{ km/s}$ is the typical interaction speed, and the cross section in this example is about $\sigma = \pi (300 \text{ AU})^2$ (from line 44). The text should thus discuss how much time it takes to get one interaction close enough to truncate the disk, and then how likely it would be to get multiple interactions.

Page 7, line 20: The text states that 200 AU disks are common. Some discussion of the distribution of observed disk sizes, along with references, should be given here. Many papers quote 100 AU as the typical disk size and find that 200 AU is more of an upper limit. The difference matters, because if most disks start out with radii of 100 AU (like HL Tau, which is discussed earlier) then they do not need to be truncated to reach 100 AU sizes...

Page 9, lines 34 to 40: The text lists two possible implications for planet formation, either planets form quickly, or they form in regions with low radiation levels. But, based on the numbers presented in the text, isn't there a third possibility? The evaporation process becomes inefficient and does not operate effectively for radial locations $r < 10 \text{ AU}$ (page 8, line 53). So why can't planets just form at radial locations in disks where $r < 10 \text{ AU}$? Planet formation models often prefer to have planets form just outside the ice-line for a number of reasons, so the best place to form planets is at about 5 AU (where evaporation is not so effective).

Page 12: On freeing floating planets:

Planet ejected from their solar systems will have a distribution of ejection speeds, where the distribution can be approximated by the form $dP/du = 4u/(1+u^2)^3$ (e.g., Moorhead and Adams 2005). As a result, if enough planets are ejected, then some fraction of them will be retained in the cluster as freely floating planets.

On the other hand, the paper by Sumi et al. (2001) (ref. [155]) has been updated by the microlensing collaboration and the estimates for the number of freely floating planets has been revised downward.

Page 13: Secondary effects: The quoted stability criterion of 'one third the binary semimajor axis' is not stringent enough. First, the paper by Holman and Weigert (Ref. [59]) only integrates over 10^4 binary orbits, which is not nearly long enough to explore long term stability, and it also corresponds to a different physical time scale for each type of binary. A better criterion is that the orbit of the planet must be less than some fraction of the binary periastron. In order to make an earth-like orbit stable for 4.6 Gyr, the age of the solar system, the binary periastron must be about 7 AU or greater (for earths orbiting the primary). So interactions that increase binary eccentricity will lead to disruption of planetary orbits. Finally, you can easily tell that 'one third of the binary semimajor axis' is not enough: If the binary has $e = 2/3$, then it can cross the planetary orbit. (And notice also that the binary eccentricity distribution $dP/de = 2e$ -- so that high-e binaries are in fact likely).

Page 15, line 30: It seems to be an oversight so discuss Planet Nine, but not cite any of the (many!) papers by Batygin and Brown. Maybe cite their recent review article (Batygin et al. 2019, Physics Report) and refer the reader to the references therein.

A general comment on the birth environment of the solar system: Many papers (like the Nice model) indicate that the giant planets formed within about 20 AU of the Sun. Since photoevaporation is inefficient that close to the Sun, isn't there a scenario where the birth cluster evaporates the solar nebula down to 25 or 30 AU, while the planets are forming at smaller radii? Note that Uranus and Neptune don't really contain any gas, so the solar nebula really just needs to retain gas at 5 to 10 AU for Jupiter and Saturn. Moreover, such a truncated solar nebula would help explain the deficit of material in the Kuiper Belt. The current text reads as if disks are either evaporated, or not, but does not leave room for partially evaporated (and thus partially truncated) disks. I have always thought that this in-between scenario was the most likely.

Page 16, Section 6, paragraph 1: While density is certainly *one* of the most important variables, I would think that the time spent in the high density environment would be on an equal footing.

Reviewer: 2

Comments to the Author(s)

Referee report for

„The birth environment of planetary systems“

by Richard J. Parker

This review article summaries very well the state-of-the-art of the research on the influence of the environment on young (forming) planetary systems. Th author presents this topic in a very structured approach that facilitates obtaining an overview of this subject. Therefore I recommend publication. However, I would suggest for the authors to consider the following points:

Introduction: Here one has the impression that the author is giving a talk to an audience of students. The style he choose might work in such a situation, however, I think the many colloquial expressions and personal reminiscences of his own thinking as student do not work out for a review article. I suggest to rephrase some of the sentences.

Throughout the author uses the term „photo-evaporation“ in the meaning of „external photo-evaporation“. For the expert this is clear, however, as this review is aimed to help newcomers in the field, it should be pointed out early on in text that photo-evaporation by the star itself and

external photo-evaporation can lead to disc destruction, but that throughout the text only external photo-evaporation is considered.

Star formation: Here I find that there are too few references to the general processes of the formation of clusters/associations and too many on issues that are of minor importance in a review article. For example, it would be useful to a novice to the field to have a reference to the typical masses and sizes of fragments (page 4, line 33), whereas he/she would be overwhelmed with 12 references to the relation between cluster mass and the mass of the most massive star, which features not very prominent in the rest of the paper.

As the authors points out the stellar density is the key parameter that determines to which degree the environment influences (forming) planetary systems. This means that differences between long- (> 50 Myr) and short- (<10 Myr)-lived stellar groups exist and that the formation, expansion of stellar groups and time scales on which this happens are essential for determining the influence of the environment. There exist different theories concerning all these points. The author tries to smooth over the various conflicts, without pointing them out. I think that newcomers should be made aware of the openness of these issues, rather than leaving them unaware of them.

Photo-evaporation: There are several groups working on the effect of external photo-evaporation. I think there is a too strong emphasis on the FRIED simulation. This should be supplemented by mentioning other works.

Eq. 3.3 and 3.4 were used nearly 20 years ago to describe the effect of external photo-evaporation, the authors themselves pointed out than that using this equations leads to overestimating the effect. Therefore, it is probably not really helpful to present them as state-of-the-art to today's students.

Minor points:

page 1 line 22. The sentence „Recent observations form ALMA suggest that planet formation may already be well under way after only 1 Myr of a star's life“ gives the, perhaps wrong, impression that all planets form within just 1 Myr. This is clearly overstating current knowledge. I think that in some cases planets might form as early as 1 Myr after a star formed reflects more the situation.

page 5 line 12: „lower-mass stars like the Sun“ is misleading as the mass of the Sun is about twice as high as the average stellar mass in a stellar group.

page 5 paragraph 3: it is not always clear whether it is referred to the local or the average density in the stellar group and references to the density are not given. same applies for paragraph 5 on the same page.

page 7 paragraph 3 and 4: the concept of the MMSN is here not very helpful because many (if not most) stars are surrounded by discs with masses below the MMSN. There are many different theories trying to explain that. Either one goes into that discussion or leaves the MMSN out altogether.

Page 7 paragraph 6: The early results were only for equal-mass stars, unfortunately, they were lateron used for other mass ratios. One should avoid that this misunderstanding continues.

page 9 paragraph 5: The dependence on the distance to the star is misleading without mentioning that it happens only on the side of the disc that is nearest to the massive star. Thus it is an asymmetric process.

page 10 argument (ii): this argument does not catch here, because one cannot generalise necessarily from our solar system to all other planetary systems

page 17 paragraph 7: The argument that the Sun formed in a short-lived stellar group (< 10 Myr) is weak as most long-lived stellar groups disperse on time scales of a few 100 Myr which is short in comparison to the Sun's age of 4.6 Gyr.

There are some incomplete sentences in the manuscript with words missing. This requires urgently changing before publication.

===PREPARING YOUR MANUSCRIPT===

===PREPARING YOUR REVISION IN SCHOLARONE===

- 1) One version identifying all the changes that have been made (for instance, in coloured highlight, in bold text, or tracked changes);
 - 2) A 'clean' version of the new manuscript that incorporates the changes made, but does not highlight them.
 - An individual file of each figure (EPS or print-quality PDF preferred [either format should be produced directly from original creation package], or original software format).
 - An editable file of each table (.doc, .docx, .xls, .xlsx, or .csv).
 - An editable file of all figure and table captions.
- Note: you may upload the figure, table, and caption files in a single Zip folder.
- Any electronic supplementary material (ESM).
 - If you are requesting a discretionary waiver for the article processing charge, the waiver form must be included at this step.
 - If you are providing image files for potential cover images, please upload these at this step, and inform the editorial office you have done so. You must hold the copyright to any image provided.
 - A copy of your point-by-point response to referees and Editors. This will expedite the preparation of your proof.

- Ensure that your data access statement meets the requirements at <https://royalsociety.org/journals/authors/author-guidelines/#data>. You should ensure that you cite the dataset in your reference list. If you have deposited data etc in the Dryad repository, please only include the 'For publication' link at this stage. You should remove the 'For review' link.
- If you are requesting an article processing charge waiver, you must select the relevant waiver option (if requesting a discretionary waiver, the form should have been uploaded at Step 3 'File upload' above).
- If you have uploaded ESM files, please ensure you follow the guidance at <https://royalsociety.org/journals/authors/author-guidelines/#supplementary-material> to include a suitable title and informative caption. An example of appropriate titling and captioning may be found at [https://figshare.com/articles/Table_S2_from_Is_there_a_trade-off_between_peak_performance_and_performance_breadth_across_temperatures_for_aerobic_sc ope_in_teleost_fishes_/3843624](https://figshare.com/articles/Table_S2_from_Is_there_a_trade-off_between_peak_performance_and_performance_breadth_across_temperatures_for_aerobic_scope_in_teleost_fishes_/3843624).

Author's Response to Decision Letter for (RSOS-201271.R0)

See Appendix A.

Decision letter (RSOS-201271.R1)

Dear Dr Parker,

It is a pleasure to accept your manuscript entitled "The birth environment of planetary systems" in its current form for publication in Royal Society Open Science. The comments of the reviewer(s) who reviewed your manuscript are included at the foot of this letter.

on behalf of Dr Mark Wyatt (Associate Editor) and Rob Ivison (Subject Editor)
openscience@royalsociety.org

Appendix A

Reviewer comments to Author:

Reviewer: 1

Comments to the Author(s)

This paper provides an updated review of how cluster environments can influence the planetary systems forming within them. The review is comprehensive and clearly written, and will provide a useful addition to the literature (especially for researchers just entering the field, including students). The paper is in good shape, and can be accepted for publication after the following relatively minor considerations have been taken into account. Although the following list is somewhat long and somewhat wordy, hopefully these issues will be straightforward to address:

The review focuses on dynamical interactions and effects due to the radiation fields provided by the cluster environment. In addition to these effects, clusters can also provide 'particle' background effects. Since the paper cannot cover everything, this omission is OK, but it should still be briefly mentioned in defining the scope of the review. The review does mention the example of short-lived radioactive nuclei in conjunction with the birth environment of the solar system (section 5) but not for the general case. In addition to their signatures in meteorites, which is rather specific to our solar system, these nuclei provide a source of ionization, which affects MRI and chemistry during the planet formation process. In addition, cluster environments can both enhance the cosmic ray flux (by trapping particles from supernovae) and shield the cosmic rays (via magnetic fields). Since the cosmic rays provide an important source of ionization, they are important for star formation (e.g., coupling the magnetic field during collapse) and planet formation (see above).

RESPONSE: I agree that these effects are interesting, but I believe fall so far out of the scope of this review that mentioning them is not necessary.

Page 3, line 33: GMCs are not really collapsing as a whole, at least not in an organized monolithic way (this finding dates back to Zuckerman and Palmer 1974). The GMCs are dynamically evolving, but 'collapsing' is not really the right description.

RESPONSE: I have re-worded this sentence.

Page 3, line 34: On a similar note, while the Jeans mass does provide a scale, saying that the cloud fragments into one solar mass pieces is overly simplistic. To see this, use the stated temperature (tens of K)

and a typical density of the molecular cloud (say 1000 particles per cc) and calculate a Jeans mass. You will find that it's larger than a typical star. You can always choose a larger density to get the answer that you want, but then you have to justify why you are [1] using a density that is larger than the typical density of the cloud, and [2] explain why you are using only a single density, instead of the rather wide range of densities sampled by the cloud. Notice also that the Jeans mass does not depend on things like angular momentum and magnetic fields, ingredients that we know are important for the star formation process. So, while Sir James Jeans is certainly correct, invoking the Jeans mass to explain stellar masses is probably more misleading than enlightening.

RESPONSE: I think this point was sloppy use of language on my part. I have edited the text to emphasise that it is the pre-stellar core distribution that peaks at around 1Msun, with stellar masses being somewhat lower.

Page 4, first full paragraph: In the discussion of stellar densities, it would be useful to point out that clusters have a density distribution, so that forming solar systems will experience a range of stellar densities. If the cluster lives long enough, then the members would sample the full distribution of stellar densities. But disks last only a few to several Myr, and planets form on the same time scale, and dynamical relaxations time are longer, so not all solar systems will fully sample the available densities in the cluster.

RESPONSE: I have added a qualifying paragraph to note that the quoted densities are the average densities.

Page 4: On a related note, while the stellar density is important for determining the degree of disruption from both dynamical interactions and radiation, the time spent in the high density environment is also important. For example, the second full paragraph (that explains the effects of different stellar densities) must be assuming something about the time spent in that environment. It's important to be more quantitative. If the time is only 1 Myr, which is on the low end for both disk lifetimes and planet formation, then the effects will be much less important than if the time is 10 Myr or 100 Myr.

RESPONSE: Most star-forming regions live for less than 10Myr, so this is the relevant timescale for the density. I have added a sentence to qualify this in the manuscript.

Page 4: Paragraph at lines 45 to 52: This paragraph asserts that the Sun formed at redshift $z=1$ when star formation (on a cosmic scale) was more active. Perhaps I am on the wrong page, but I find that the

redshift for the formation of the Sun is more like $z=0.43$ (although the result depends on what cosmological parameters are assumed). Perhaps the author is confusing the age of the universe and the look back time? The age of the universe was about 4 Gyr at redshift $z=1$, but that was 9 Gyr ago, about twice the age of the solar system. In any case, the cosmology used to get this result should be specified. And if $z=0.43$, as I suspect, then the rest of the paragraph is less relevant.

RESPONSE: I have re-worded this slightly. The point is that the exoplanet host stars are older than stars in SF regions today, and may have formed in high density regions.

Page 5: Equation (2.2) seems to have an error/typo: Maybe the cluster mass should be on top? Even in that case, if one uses a cluster mass $M_c = 2000 M_{\text{sun}}$ (like the ONC) and a galaxy mass $M_g = 10^{10} M_{\text{sun}}$, and $R_G = 8 \text{ kpc}$ (like the solar circle), then the Jacobi radius $r_J = 47 \text{ pc}$. The radius of the ONC is usually quoted as $r = 2 \text{ pc}$ (e.g., Hillenbrand and Hartmann 1997). Which leads to the question: How does an outer boundary at 47 pc (from external galactic tides) affect a star cluster with a radius of only 2 pc?

RESPONSE: The typo in the equation has been corrected. If one uses the actual mass of the Galaxy in this equation ($10^{12} M_{\text{sun}}$), then the Jacobi radius for a $1000 M_{\text{sun}}$ cluster is 8 pc. The Jacobi radius is much lower for a low-mass cluster, so entirely relevant to the discussion topic as a star cluster's half-mass radius increases by a factor of several during dynamical relaxation. I have added numbers to quantify this statement in the text.

Page 6, line 37: Although the review focuses on dynamical interactions and evaporation from the cluster radiation, the disks also lose mass by disk accretion onto the star, additional evaporation by stellar photons, and even by locking up some mass into their planets. For completeness, these channels should be mentioned, and this point in the text (right before section 3a) would be a good place.

RESPONSE: Agreed, and I have added text to the relevant paragraph.

Page 6, line 52: The text says that stars can undergo multiple interactions. It would be useful to add some further explanation to clarify the conditions required to get multiple interactions. The interaction rate is (approximately) $\Gamma = n \sigma v$ where $n = 100 \text{ pc}^{-3}$ is the stellar density, $v = 1 \text{ km/s}$ is the typical interaction speed, and the cross section in this example is about $\sigma = \pi (300 \text{ AU})^2$ (from line 44). The text should thus discuss how much time it takes to get one interaction close enough to truncate the disk, and

then how likely it would be to get multiple interactions.

RESPONSE: This discussion is better placed later on, where I already discuss the cross section for interactions in Section 4.

Page 7, line 20: The text states that 200 AU disks are common. Some discussion of the distribution of observed disk sizes, along with references, should be given here. Many papers quote 100 AU as the typical disk size and find that 200 AU is more of an upper limit. The difference matters, because if most disks start out with radii of 100 AU (like HL Tau, which is discussed earlier) then they do not need to be truncated to reach 100 AU sizes...

RESPONSE: I have added relevant references.

Page 9, lines 34 to 40: The text lists two possible implications for planet formation, either planets form quickly, or they form in regions with low radiation levels. But, based on the numbers presented in the text, isn't there a third possibility? The evaporation process becomes inefficient and does not operate effectively for radial locations $r < 10$ AU (page 8, line 53). So why can't planets just form at radial locations in disks where $r < 10$ AU? Planet formation models often prefer to have planets form just outside the ice-line for a number of reasons, so the best place to form planets is at about 5 AU (where evaporation is not so effective).

RESPONSE: Agreed, some text has been added to clarify point (i).

Page 12: On freeing floating planets:

Planet ejected from their solar systems will have a distribution of ejection speeds, where the distribution can be approximated by the form $dP/du = 4u/(1+u^2)^3$ (e.g., Moorhead and Adams 2005). As a result, if enough planets are ejected, then some fraction of them will be retained in the cluster as freely floating planets.

RESPONSE: Agreed, and the fact that some free-floating planets can be retained by the cluster is already mentioned in the text.

On the other hand, the paper by Sumi et al. (2001) (ref. [155]) has been updated by the microlensing collaboration and the estimates for the number of freely floating planets has been revised downward.

RESPONSE: I have added a reference to the more recent paper by Mroz et al 2017.

Page 13: Secondary effects: The quoted stability criterion of 'one third the binary semimajor axis' is not stringent enough. First, the paper by Holman and Weigert (Ref. [59]) only integrates over 10^4 binary orbits, which is not nearly long enough to explore long term stability, and it also corresponds to a different physical time scale for each type of binary. A better criterion is that the orbit of the planet must be less than some fraction of the binary periastron. In order to make an earth-like orbit stable for 4.6 Gyr, the age of the solar system, the binary periastron must be about 7 AU or greater (for earths orbiting the primary). So interactions that increase binary eccentricity will lead to disruption of planetary orbits. Finally, you can easily tell that 'one third of the binary semimajor axis' is not enough: If the binary has $e = 2/3$, then it can cross the planetary orbit. (And notice also that the binary eccentricity distribution $dP/de = 2e$ -- so that high-e binaries are in fact likely).

RESPONSE: The sentence starts 'Typically,...' but I have added further qualification. I don't know where the referee's value of 7au for the Earth comes from. However, the binary eccentricity distribution does not go as $dP/de = 2e$ - see Duchene & Kraus 2013 - so high eccentricity binaries are not more common than low-eccentricity systems.

Page 15, line 30: It seems to be an oversight so discuss Planet Nine, but not cite any of the (many!) papers by Batygin and Brown. Maybe cite their recent review article (Batygin et al. 2019, Physics Report) and refer the reader to the references therein.

RESPONSE: This was a typo - I had intended to cite the Brown and Batygin 2016 paper. I have also cited the Batygin et al 2019 review, as requested.

A general comment on the birth environment of the solar system: Many papers (like the Nice model) indicate that the giant planets formed within about 20 AU of the Sun. Since photoevaporation is inefficient that close to the Sun, isn't there a scenario where the birth cluster evaporates the solar nebula down to 25 or 30 AU, while the planets are forming at smaller radii? Note that Uranus and Neptune don't really contain any gas, so the solar nebula really just needs to retain gas at 5 to 10 AU for Jupiter and Saturn. Moreover, such a truncated solar nebula would help explain the deficit of material in the Kuiper Belt. The current text reads as if disks are either evaporated, or not, but does not leave room for partially evaporated (and thus partially truncated) disks. I have always thought that this in-between scenario was the most likely.

RESPONSE: I have added a qualifying statement to this effect.

Page 16, Section 6, paragraph 1: While density is certainly *one* of the most important variables, I would think that the time spent in the high density environment would be on an equal footing.

RESPONSE: Most star-forming regions live for less than 10Myr before dissolving into the Galactic field, so the time spent in a dense environment is largely constant for planetary systems (as addressed in my response to one of the referee's earlier points).

Reviewer: 2
Comments to the Author(s)

Referee report for

„The birth environment of planetary systems“

by Richard J. Parker

This review article summaries very well the state-of-the-art of the research on the influence of the environment on young (forming) planetary systems. The author presents this topic in a very structured approach that facilitates obtaining an overview of this subject. Therefore I recommend publication. However, I would suggest for the authors to consider the following points:

Introduction: Here one has the impression that the author is giving a talk to an audience of students. The style he choose might work in such a situation, however, I think the many colloquial expressions and personal reminiscences of his own thinking as student do not work out for a review article. I suggest to rephrase some of the sentences.

RESPONSE: I was trying to do something different here. I have consulted several colleagues about this and all of them, whilst recognising it's quite an unconventional introduction, think it works well and so I would like to keep it as it is.

Throughout the author uses the term „photo-evaporation“ in the meaning of „external photo-evaporation“. For the expert this is clear, however, as this review is aimed to help newcomers in the field, it should be pointed out early on in text that photo-evaporation by the star itself and external photo-evaporation can lead to disc destruction, but that throughout the text only external photo-evaporation is considered.

RESPONSE: I have added a qualifying statement just before the start of Section 3a.

Star formation: Here I find that there are too few references to the general

processes of the formation of clusters/associations and too many on issues that are of minor importance in a review article. For example, it would be useful to a novice to the field to have a reference to the typical masses and sizes of fragments (page 4, line 33), whereas he/she would be overwhelmed with 12 references to the relation between cluster mass and the mass of the most massive star, which features not very prominent in the rest of the paper.

RESPONSE: I have added some further relevant references here.

As the authors points out the stellar density is the key parameter that determines to which degree the environment influences (forming) planetary systems. This means that differences between long- (> 50 Myr) and short- (<10 Myr)-lived stellar groups exist and that the formation, expansion of stellar groups and time scales on which this happens are essential for determining the influence of the environment. There exist different theories concerning all these points. The author tries to smooth over the various conflicts, without pointing them out. I think that newcomers should be made aware of the openness of these issues, rather than leaving them unaware of them.

RESPONSE: It is beyond the scope of the review to go into details on this, but my other responses to this reviewer and the other reviewer's queries on the timescale issue should address this point.

Photo-evaporation: There are several groups working on the effect of external photo-evaporation. I think there is a too strong emphasis on the FRIED simulation. This should be supplemented by mentioning other works.

RESPONSE: The earlier work in this area is cited. However, I'm struggling to think of a current rival to the FRIED grid.

Eq. 3.3 and 3.4 were used nearly 20 years ago to describe the effect of external photo-evaporation, the authors themselves pointed out than that using this equations leads to overestimating the effect. Therefore, it is probably not really helpful to present them as state-of-the-art to today's students.

RESPONSE: Equation 3.4 is still the de facto state of the art to describe EUV photoevaporation (see recent papers by Winter et al), and rather than using the FRIED grid (which would put the review into the realm of primary, or new research), I thought it better to mention the FRIED grid but use the older FUV prescription.

Minor points:

page 1 line 22. The sentence „Recent observations form ALMA suggest that planet formation may already be well under way after only 1 Myr of a star's

life“ gives the, perhaps wrong, impression that all planets form within just 1 Myr. This is clearly overstating current knowledge. I think that in some cases planets might form as early as 1 Myr after a star formed reflects more the situation.

RESPONSE: I have removed the word 'well', to avoid the impression that planet formation is complete after such short timescales.

page 5 line 12: „lower-mass stars like the Sun“ is misleading as the mass of the Sun is about twice as high as the average stellar mass in a stellar group.

RESPONSE: I have altered the text accordingly.

page 5 paragraph 3: it is not always clear whether it is referred to the local or the average density in the stellar group and references to the density are not given. same applies for paragraph 5 on the same page.

RESPONSE: I have added some text to qualify this (based also on the other referee's comments). I have made it clear that I am referring to the average density, but that there is a range.

page 7 paragraph 3 and 4: the concept of the MMSN is here not very helpful because many (if not most) stars are surrounded by discs with masses below the MMSN. There are many different theories trying to explain that. Either one goes into that discussion or leaves the MMSN out altogether.

RESPONSE: Agreed, but the discussion relates to the masses of solids in the Solar system, and I don't extrapolate to discs around other stars.

Page 7 paragraph 6: The early results were only for equal-mass stars, unfortunately, they were later on used for other mass ratios. One should avoid that this misunderstanding continues.

RESPONSE: I'm not sure what the referee is referring to here.

page 9 paragraph 5: The dependence on the distance to the star is misleading without mentioning that it happens only on the side of the disc that is nearest to the massive star. Thus it is an asymmetric process.

RESPONSE: I disagree. What is more misleading is the impression given by some authors that the disc-bearing stars are fixed in space with respect to the high mass stars, which is incorrect as the stars move around on order of a crossing time (usually ~ 0.1 Myr in a young, dense star-forming region).

page 10 argument (ii): this argument does not catch here, because one cannot generalise necessarily from our solar system to all other planetary

systems

RESPONSE: If massive stars preclude gas giant planet formation, but if a planetary system with gas giants contains evidence for SLRs (indicating proximity to massive stars), then I don't see how the argument can NOT be generalised to all planetary systems.

page 17 paragraph 7: The argument that the Sun formed in a short-lived stellar group (< 10 Myr) is weak as most long-lived stellar groups disperse on time scales of a few 100 Myr which is short in comparison to the Sun's age of 4.6 Gyr.

There are some incomplete sentences in the manuscript with words missing. This requires urgently changing before publication.

RESPONSE: I have fixed these typos.